# The Universal Cloud and Aerosol Sounding System (UCASS): a low-cost miniature optical particle counter for use in dropsonde or balloon-borne sounding systems.

Helen R. Smith[1], Zbigniew Ulanowski[1], Paul H. Kaye[1], Edwin Hirst[1], Warren Stanley[1], Richard Kaye[1], Chris Stopford[1], Andreas Wieser[2], Maria Kezoudi[1], Joseph Girdwood[1], Richard Greenaway[1], and Robert MacKenzie[1]

[1]Centre for Atmospheric and Climate Physics Research, School of Physics, Astronomy and Maths, University of Hertfordshire, Hatfield, Hertfordshire, AL10 9AB
[2]Institute of Meteorology and Climate Research, Karlsruhe Institute of Technology (KIT), 76021 Karlsruhe, Germany.

**Correspondence:** Zbigniew Ulanowski (z.ulanowski@herts.ac.uk)

**Abstract.** A low-cost miniaturized particle counter has been developed by The University of Hertfordshire (UH) for the measurement of aerosol/droplet concentrations and size distributions. The Universal Cloud and Aerosol Sounding System (UCASS) is an Optical Particle Counter (OPC), which uses wide-angle elastic light scattering for the high precision sizing of fluid-borne particulates. The UCASS has up to 16 configurable size bins, capable of sizing particles in the range 0.4–40 μm diameter. Unlike traditional particle counters, the UCASS is an open-geometry system which relies on an external air flow. Therefore the instrument is suited for use as part of a dropsonde, balloon-borne sounding system, as part of an Unmanned Aerial Vehicle (UAV), or on any measurement platform with a known air flow. Data can be logged autonomously using an on-board SD card, or the device can be interfaced with commercially available meteorological sondes to transmit data in real time. The device has been deployed on various research platforms to take measurements of both droplets and dry aerosol particles. Comparative results with co-located instrumentation in both laboratory and field settings are used to assess the performance of the UCASS, and illustrate potential uses.

*Copyright statement.* TEXT

## 1 Introduction

Atmospheric aerosols are a key component in the Earth's radiative system as they modify the local and planetary albedo by way of direct and indirect effects. Aerosols directly impact the radiation budget via the scattering and absorption of solar radiation, and to a lesser extent, the scattering, absorption and emission of terrestrial radiation (Li et al., 2010; Zhou and Savijärvi, 2014). At the Top Of Atmosphere (TOA), radiative forcing due to direct aerosol effects is estimated at -0.35(-0.85 to +0.15) W m$^{-2}$ (Myhre et al., 2013). Indirect effects arise from aerosols acting as Cloud Condensation Nuclei (CCN) or Ice Nuclei (IN), thus influencing the formation and evolution of clouds (Twomey, 1977). Measurements from satellites observe the Cloud Radiative

Effect (CRE) to be -50 W m$^{-2}$ in the short-wave and +30 W m$^{-2}$ in the long-wave, with estimates varying by 10 % (Loeb et al., 2009). The significant uncertainties of both direct and indirect effects, combined with the subsequent rapid adjustments and feedbacks, prompted the 2013 Intergovernmental Panel on Climate Change (IPCC) to cite Aerosol-Radiation Effects (ARE) and Aerosol-Cloud interactions as the largest sources of uncertainty in predicting climate change today (IPCC2013, 2013).

The uncertainties stem from a range of causes, including the inadequate characterisation of aerosol optical properties, the inaccurate representation of their spatial and temporal coverage, and the complex nature of interactions between cloud and aerosol (Ramanathan et al., 2001; Twomey, 1977; Charlson et al., 2001; Storelvmo, 2012).

To constrain uncertainties, a varied approach to aerosol measurement is required. To cover large geographical scales, there exists a multitude of measurements from ground based sun photometer networks (Holben et al., 1998; Che et al., 2009; Bokoye

et al., 2002) to satellite based instruments (MODIS, MISR, POLDER, PARASOL)(Huete, 2004; Chu et al., 2003; Kaufman et al., 1997; Zhang and Christopher, 2003; Jiao et al., 2018). This combination of instruments provide near-continuous measurements world-wide and yield retrievals of aerosol properties such as Aerosol Optical Depth (AOD), number concentration and size distribution. This large-scale coverage is crucial in capturing the spatial extent of atmospheric aerosol, and the high temporal resolution permits the monitoring of long-term diurnal, seasonal and annual trends. However, there exist considerable

discrepancies between satellite products due to uncertainties in calibration, assumed aerosol microphysics, sampling and cloud screening (Kokhanovsky et al., 2010).

Many other model studies have demonstrated the sensitivity of ARE to the aerosol layer height (Vuolo et al., 2014; Mishra et al., 2015), thus highlighting the need to faithfully represent the the vertical distribution of aerosol in global models which remains a serious challenge. Therefore, in-situ studies are necessary for obtaining high-quality, comprehensive datasets for the

20 microphysical characterization of aerosol, the testing of retrieval algorithms and the quality assurance of remote sensing data. Vertically resolved in-situ data are typically gathered with aircraft based instrumentation during research campaigns. These campaigns employ a variety of instruments for particle measurement: from single-scattering particle probes for the counting and sizing of small particles; optical array probes for the imaging of larger particles; and filters to collect samples for in-depth chemical analysis. Despite this assemblage of techniques, there remain deficiencies in measurement capabilities. Aircraft

campaigns are expensive and subject to accessibility issues, thus limiting the time, space and location each campaign may cover. Whilst co-located remote and in-situ measurement campaigns are invaluable for the validation of retrieval algorithms, the geographical extent of in-situ measurements remains limited. Alternative height-resolved in-situ measurement techniques are required to bridge this gap between the comprehensive, but spatially limited aircraft campaigns, and the expansive remote sensing networks. Progress has been made in recent years through the miniaturization of instrumentation. The reduction in

weight and size allow instruments to be used on alternative platforms such as weather balloons and UAVs, and as such, various miniature Optical Particle Counters (OPCs) for balloon based studies have been developed. For the small size ranges, the Printed Optical Particle Spectrometer (POPS) (Gao et al., 2016) utilizes a blue laser for the counting and sizing of particles from 140–3000 nm, making it suitable for PM2.5 measurements. For larger particles, the Cloud Particle Sensor (CPS) uses a 790 nm laser for the sizing of particles between 2 and 80 μm (Fujiwara et al., 2016). The Light Optical Aerosol Counter

(LOAC) covers a wide size range from 0.2–100 μm using a 650 nm laser. The LOAC takes measurements of scattered intensity

at two angular ranges, which are used to determine size and estimate the refractive index of the scattering particle (Renard et al., 2016).

This paper discusses a novel instrument, the Universal Cloud and Aerosol Sounding System (UCASS): a low-cost, lightweight (280 g), open path Optical Particle Counter (OPC), designed for use as a dropsonde or as part of a balloon-borne sounding system. Whilst other balloon-based instrumentation exists, the UCASS is the first (to the best of our knowledge) particle counter designed for dropsonde based systems. Routine meteorological soundings are performed daily at a number of research stations worldwide, gathering information about temperature, pressure, humidity and wind. These radiosoundings can be combined with further instrumentation to measure additional data such as ozone distribution and electric field (Jenkins et al., 2015; Nicoll, 2012). However these soundings do not currently take measurements of atmospheric particles. The UCASS offers the ability to incorporate particle measurements into these soundings, giving vertical distributions of aerosol/droplet concentrations and size distributions. The UCASS may be used as both a dropsonde or upsonde and can therefore provide complimentary data to aircraft based campaigns. The use as a dropsonde is of particular benefit as the aircraft can drop the sondes through a specific airmass, allowing for more targetted measurements. Furthermore, due to the relative ease and affordability of radiosoundings, the UCASS also offers an alternative to aircraft based measurements with fewer time and space restrictions.

## 2   Instrument Design

### 2.1   Assembly and optical set-up

For ease of reading, the following section defines a common coordinate system for figures 1–4. Furthermore, we define a common set of identifiers, where alphanumeric labelling is used to identify physical parts, and roman numerals are used to identify scenarios or alternative views (within the coordinate system). The structural components of the UCASS are made primarily from 3D printed parts as shown in Fig. 1. The design is essentially tubular, 180 mm long and 64 mm in diameter. An elliptical cross-section hole, 30 mm × 22 mm in size, runs the full length of the UCASS body and offers a low impedance path for the sampled aerosol, here, the air flow travels in the $y$ direction. The UCASS is designed with this particular shape for use as a dropsonde system, whereby a connected KITsonde fits inside the UCASS, and the entire payload fits within a standard release container. This is discussed further in Sect. 3.2.1. The main unit (a1) is 3D printed from nylon using Selective Laser Sintering (SLS). This process is chosen for the main unit as it produces mechanically rigid and thermally stable chassis, which is crucial for the instrument to endure the temperature cycle associated with a routine sounding. The process uses a layer thickness of 0.12 mm, which is adequate for the placement of mechanical components such as batteries and circuitry. However, to improve the precision for optical components, a modular design is used, thus allowing each optical element to be aligned and secured individually. The beam forming optics are mounted onto an insert (a2), which is 3D printed from Polyactic Acid (PLA) using Fused Filament Fabrication (FFF). This method is chosen for the increased mechanical rigidity which is required for the thin wall thicknesses. and the elliptical mirror is mounted on aluminium insert (a3), which is machined using a 5-axis mill. Once the beam forming optics are aligned, the insert (a2) is secured into place in the main unit. The elliptical

mirror, mounted on insert (a3) is aligned using three adjustment screws which secure the mirror onto the main unit (a1). One assembled, the UCASS is enclosed by a 3D printed sleeve (a4) and (a5).

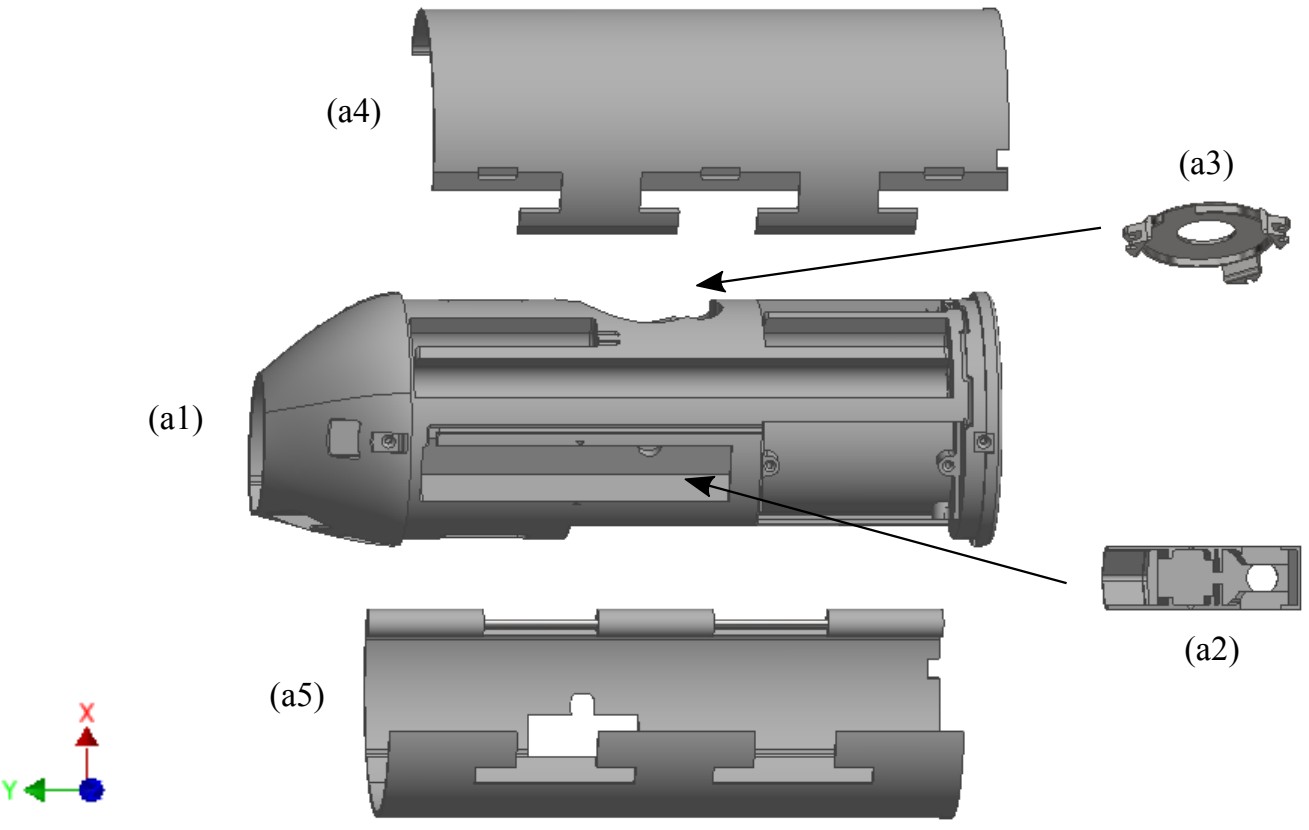

**Figure 1.** Structural components of the UCASS. This consists of 4 3D printed components: the main housing unit (a1), inserts for the beam forming optics (a2) and the outer casing (a4) and (a5). The mirror holder (a3) is machined out of aluminium.

The optical assembly is shown in Fig. 2. Here, the UCASS chassis (a1) is shown as a transparent layer to illustrate the placement of the optical elements. The top panel shows a view along the $xy$ plane (parallel to the airflow), and the bottom panel shows a view in the $xz$ plane (perpendicular to the airflow). The input beam is a 658 nm continuous-wave diode laser (b1) from Oclaro (part number HL6501MG), operating at 10 mW. The laser is fitted with a collimator from the Optoelectronics Company (part number 500-020012). The collimated beam of elliptical cross-section approximately 3.5 mm (long axis) by 2 mm (short axis) is focussed along the short axis by a 50mm focal length cylindrical lens (b2) from Edmund Optics (part number 68-047) and then passes through a 2 mm aperture (b3) perpendicular to the beam's long axis. For ease of assembly, these beam-forming components (parts (b1), (b2), and (b3) in Fig. 2) are mounted onto an insert (part (a2) in Fig. 1). A plane 9 mm × 9 mm front silvered mirror (b4) from Edmund optics (part number 31-004) reflects the beam at 45 °, directing it

across the flow of air through the UCASS body, it is then extinguished within the UCASS casing. The input beam is shown in red on Fig. 2. The 2 mm aperture has the effect of reforming the beam's Gaussian intensity profile into an approximate 'top hat' profile such that, at the sensing area (coincident with the focal distance of the cylindrical lens), the beam has a relatively uniform intensity cross-section of $\approx$2 mm width by $\approx$40 μm depth. Particles passing through the beam parallel to its short axis therefore experience similar levels of irradiance, a prerequisite for accurate particle sizing. This is discussed in Sect. 2.2.

Particles carried in the airflow through the UCASS may pass through the laser at any point in its traverse across the air flow path. However, only those passing through a specific 0.5 mm$^2$ sensing area within the laser beam are measured. This sensing area is defined optically by a custom designed combination of concave elliptical mirror (b5) and a dual-area photodiode detector (b6). Both mirror and detector were designed by the University of Hertfordshire and are now commercially produced by Alphasense Ltd (part number 836-0001-00) and First Sensor GmbH (part number DP12.5-6 SMD) respectively, available only through special order. The elliptical mirror (b5) collects light scattered by a particle within the sensing area of the laser beam within a solid angle element of 1.69 sr (scattering angles from 16 ° to 104 °) and directs it towards the photodiode detector (b6), where the total intensity is measured.

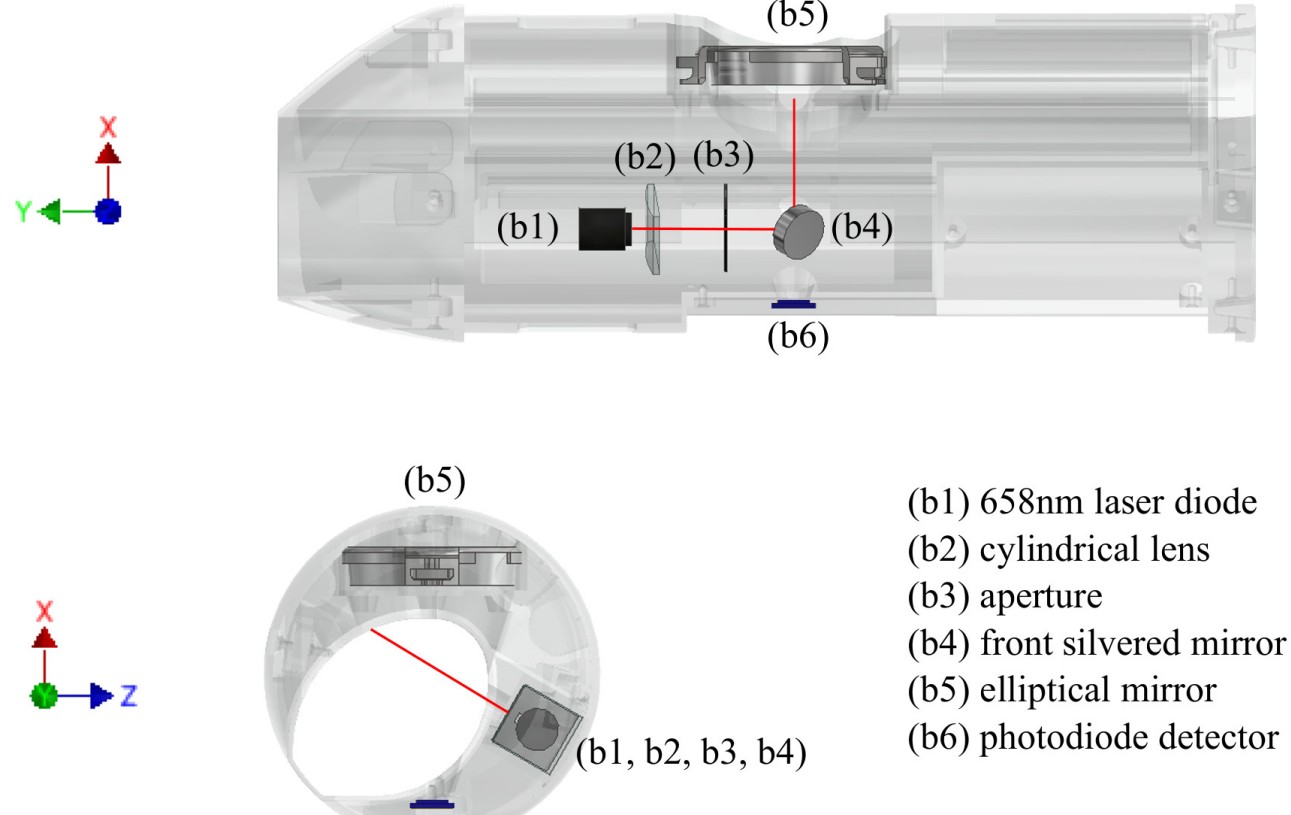

**Figure 2.** Optical assembly of the UCASS. The top panel shows a view in the $xy$ plane, parallel to the airflow. The the bottom panel shows a view in the $xz$ plane, perpendicular to the airflow. The beam forming optics consist of a laser with collimator (b1), a cylindrical lens (b2) and a 2 mm aperture (b3). The beam is directed into the instrument via a front silvered mirror (b4), angled at 45 ° to the beam. Particles that cross the laser beam will scatter light. The elliptical mirror (b5) collects light scattered between angles 16 ° and 104 ° and focuses this onto the detector (b6), where the pulse height and duration is recorded.

## 2.2 Particle detection and defining the sensing area

The photodiode detector, shown in Fig. 3(i), is a dual area design comprising two electrically isolated photosensitive regions - a central rectangular region measuring 1.9 mm $\times$ 1.0 mm surrounded by a circular region of diameter 4 mm. The design is such that, particles which are contained centrally within the sensing area will cause scattered light to be focused onto the inner detector only, whereas particles passing immediately outside of the sensing area will cause the scattered light to be focused onto the outer detector - therefore allowing the sample area to be defined optically.

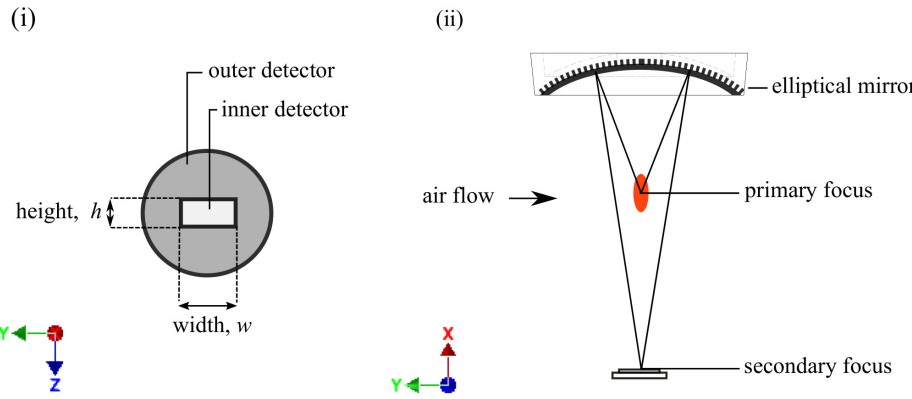

**Figure 3.** (i) The dual element detector consists of an inner rectangular detector of width $w$ and height $h$. (ii) A view of the optical system orthogonal to the air flow: the laser beam is centred on the primary focus of the elliptical mirror, and the the detector is centred at the secondary focus. The width, $w$, of the inner detector dictates the length of the laser beam (along the $z$ axis, orthogonal to the air flow) in which a particle can be detected. The height, $h$, of the inner detector is wider than the path of a particle crossing the laser beam, and therefore allows the entire pulse caused by a particle transit to be recorded.

The geometry of the detector, mirror and laser are used to define the sensing area. As shown in Fig. 3(ii), the focal point of the laser beam is centred at the primary focus of the elliptical mirror, and the detector is centred at the secondary focus. The inner detection area is a rectangle, where the width , $w$, directly governs the length of the laser beam (along the beam axis) in which a particle can be detected - and therefore defines one dimension of the sensing area. Figure 4 shows a rotated view of Fig. 3(ii), this time viewed in the direction of the air flow (the $y$ axis). The image is also rotated slightly around the $z$ axis in order to show the image formed on the detector Figures 4(i), (ii) and (iii), show how the width, $w$, of the inner detector governs the length of the sensing area in the $z$ axis. In this direction, particles passing through the primary focus of the elliptical mirror will cause scattered light to be focused at the secondary focus. However, only those passing within $\pm\frac{1}{2}w$ of the centre of the sensing area will be focused onto the inner detector, whereas particles passing to the left or right will cause light to be focused onto the outer detector (as shown in Fig. 4(iii) and will not be counted. Figure 4(iv) and (v) shows how the focus of the elliptical mirror is used to define the dimension of the sensing area along the $y$ axis. Particles passing above or below the primary focus of the mirror will cause scattered light to be focused above or below the detector - creating an enlarged image to form on the detector. In the current arrangement, particles passing 0.15 mm above/below the primary focus will create an image large enough to illuminate both the inner and outer detector, as shown in Fig. 4(v), at which point the particle is considered to be outside of the sensing area. This distance is governed by the geometry of the mirror and the size of the inner detector. For particles straddling the boundary of the sensing volume, light will be focused onto both the inner and outer detectors and therefore a criteria is defined to determine when the particle should or should not be counted. Based upon the intensity measured by the inner detector ($I_1$), and the intensity measured by the outer detector ($I_2$): if $I_2 \leq \frac{1}{4}I_1$ then the particle is considered to be inside the sensing area. The airflow is perpendicular to the sensing area, and as such, all particles passing through the sensing area (occupying the $xz$

plane) will traverse the depth of the laser beam (in the $y$ direction). The depth of the laser beam dictates the length of time the particle spends in the beam, but does not dictate the number of particles sampled. We therefore define sample area, and not sample volume. The total volume of sampled air is then calculated from the sensing area and the speed of the airflow.

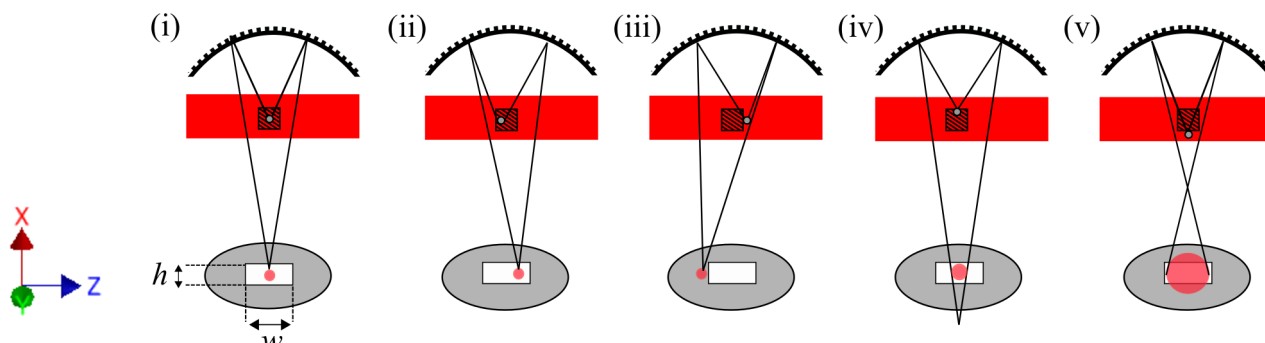

**Figure 4.** Optically defining the sensing area. The sensing area is shown as a shaded square within the laser beam, particles are represented by grey dots. The airflow direction is in the $y$ direction. (i) particles passing through the primary focus in the centre of the sensing area cause scattered light to be focussed at the centre of the detector, (ii) particles passing to the left/right of the primary focus, but still within the sensing area, will cause scattered light to be focussed to the right/left of the centre of the inner detector, (iii) particles passing more than $\frac{1}{2}w$ to the left/right of the centre of the sensing area will cause scattered light to be focussed onto the outer detector, (iv) particles passing above/below the primary focus will cause the scattered light to be focussed below/above the secondary focus and therefore below/above the detector, resulting in a larger image on the detector, (v) particles passing more than 0.15 mm above/below the primary focus will cause scattered light to fall on both the inner and outer detector; the ratio of the inner and outer detector signals are used to determine whether particles are are considered to be in the sensing area.

Particles passing through the sensing area will pass through the short axis of the focussed laser beam (in the $y$ direction, causing a pulse of light to fall on the detector. The width of this pulse is equal to the Time-of-Flight (ToF) of the particle across the beam, and the height of this pulse will correspond to the maximum intensity incident upon the detector during this transition. The pulse height depends upon the amount of light scattered by the particle: a function of particle size, optical properties and the intensity of the incident light. Therefore, the pulse height is used to derive particle size, whilst ToF data is stored for data quality assurance. To ensure that the laser beam intensity is consistent across the sample area, the beam is shaped with the use of a 2 mm aperture, thus minimising sizing errors due to beam non-uniformity. Figure 5 shows the relative intensity distributions across the major (left) and minor (right) axes. As discussed above, particles passing at the edges of the major axis will form unfocussed images on the detector and therefore will not be counted, only the central 0.3 mm section of the major axis falls within the sensing area.

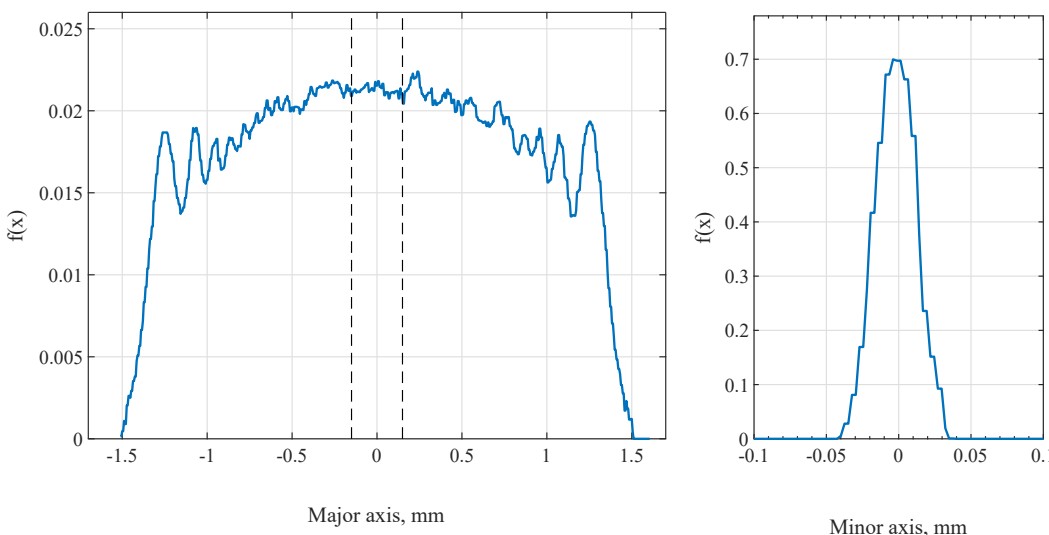

**Figure 5.** Probability density functions $f(x)$ of the laser intensity across the major (left) and minor (right) axes of the focussed laser beam. Only particles passing within the central 0.3 mm of the major axis (shown be dashed lines) will be considered withing the sensing area, and thus only these will be counted.

### 2.2.1 Stray Light Considerations

Because the UCASS is an open geometry instrument, the system is subject to stray light when operated during daylight. This is issue is counteracted with a two-fold approach. Firstly, the interior of the instrument is coated with a highly absorbing paint (Stuart Semple Black 2.0) which reduces the amount of sunlight reflected down the inlet. Whilst many light-absorbing coatings are highly directional, this paint exhibits high absorbency for direct and glancing angles across the visible and near-infra red wavelength range (the sensitivity range of the detector). This paint drastically reduces the amount of sunlight incident on the detector through reflection, however, due to the geometry of the system, sunlight may still fall directly incident on the detector

for angles <20°. To account for the remaining stray light, he on-board electronics dynamically monitors the background signal. This background signal is removed from each pulse prior to saving, thus removing the effect of stray light.

## 2.3 Electronics

A PIC microcontroller (PIC18F27J53) based electronic circuit governs the operation of the UCASS and how measurements
using the dual-photodiode system are made and recorded. A simplified block diagram of the electronics system is shown in Fig. 6. The photodiode sensor consists of two photosensitive regions, the inner detector and the outer detector. A particle within the sensing area will cause some scattered light to fall incident on the inner detector, and so this inner detector is used to create a trigger. The central element is connected to a circuit that converts the photo current to a voltage (0–3.3 VDC). The background signal is also monitored in order to account for stray light, as discussed in Sect. 2.2.1. The background signal is removed
from the measured photocurrent, a process denoted here as 'DC restoration'. The remaining signal will activate a trigger if the voltage amplitude detected goes above a pre-defined threshold ($\approx$5 mV). In this event, the trigger comparator determines that there is a particle in the sensing area, and notifies the core microcontroller. The microcontroller then activates the peak-detector electronics that will sample and hold the peak of the light signal on both the inner and outer detectors as the particle passes through the laser beam. Before saving the peak value, a Inner/Outer signal ratio comparator is used to determine whether the
particle is wholly within the sample area (as discussed in Sect. 2.2). If the particle is deemed to be within the sample area, then the microcontroller then digitises both signals with an on-board 12bit Analog-to-Digital Converter (ADC) and categorises the combined peak signal into one of 4095 bins dependent upon the value of the amplitude. If the pulse height is too high, the detector will saturate and the particle will not be counted. This would be the case for the upper limit of the detectable size range. If the pulse is too low, the peak signal will not be sufficient to trigger a measurement, and the pulse will not be recorded.
This would be the case when the particle is below the threshold of the measurable size range.

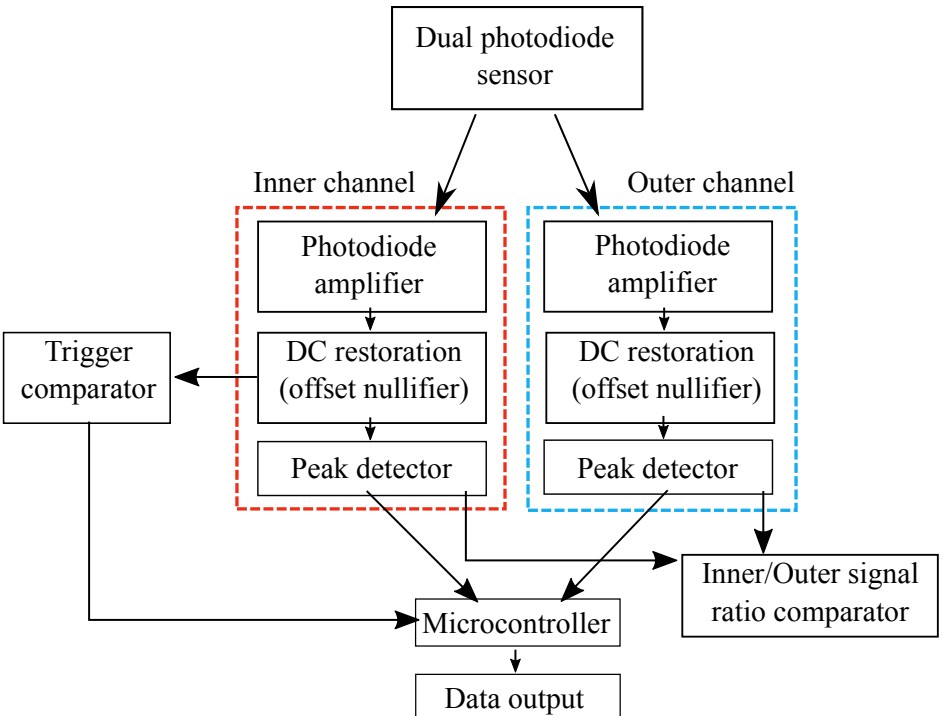

**Figure 6.** Particles passing through the sensing area will cause light to fall on the inner element of the photodiode sensor. To account for stray light, the background signal is removed, a process labelled here as DC restoration. If the remaining signal on the inner channel goes above a threshold of 5 mV, the trigger comparator notifies the microcontroller that there is a particle within the sensing area. The signals inner and outer channels are then measured, to determine the total light incident on the detector (again, the background signal is removed). The Inner/Outer signal ratio comparator determines whether the particle is wholly inside the sensing area (as discussed in Sect. 2.2), and if the particle is deemed inside the sensing area, the data is saved.

The measurable size range is determined by the laser power, amplifier gain and detector sensitivity. The hardware can be modified by changing resistor values in the electronic circuit, thus altering the amplifier gain. Therefore, two versions of the UCASS hardware are available: the 'high-gain' version with a nominal measurable size range of of 0.4–17μm, and the 'low-gain' version with a nominal measurable size range of 1–40μm.

5      Particle-by-Particle (PbP) pulse height recording is utilised for the calibration (discussed in section 2.4.2, this utilises all 4095 bins allowing the digitised pulse for each particle to be recorded individually. However, during routine measurements, the data are compressed into fewer size bins, with thresholds & limits defined by the user. Measurements are rejected if the particle is deemed to be outside of the sample area (as discussed in Sect. 2.2), measurements are also rejected if the ToF is deemed too short or too long. A short ToF would occur if an event such as electrical noise spikes triggered the system. A long ToF

10      would be the case if a large body or agglomeration of particles passed through the beam. The default ToF minimum/maximum limits are 1 and 100 μs, respectively. As the ToF is a function of particle size (amongst other factors), particles may produce legitimate ToFs above/below the rejection criteria. However, as the UCASS can only measure particles within a specific size

range, the ToF rejection criteria are chosen such that particles within the measureable size range cannot realistically produce ToFs outside of the ToF criteria.

In this manner, the microcontroller continuously assembles histograms of particle size/count data. Means of ToF data are also recorded for a subset of the size bins to allow some use of this parameter to monitor flow rate through the device. At every one-second interval, the histogram data-set is either saved to an on-board micro SD card or transmitted via a serial link (XDATA protocol) to a radiosonde device for Radio Frequency (RF) transmission of the data. The histogram data are cleared following each data save/transmit event. The XDATA protocol is limiting in terms of bandwidth such that there is only space to utilise 10 of the available 16 size bins. The configuration of the bin boundaries can be altered to accommodate this and make best use of the bin quantity available. The device electronics can measure up to $10^4$ particles per second and can operate in air flow speeds between 2 and 15 ms$^{-1}$ with the standard firmware. For a standard operating velocity of 5 ms$^{-1}$, this equates to a particle concentration of $3.5 \times 10^9$ m$^{-3}$. This is roughly an order of magnitude below the concentration where coincidence errors are likely to become problematic. The firmware can be modified to change the ToF window, thus allowing shorter or longer particle transitions to be recognised if the measurements platform requires it (this is discussed further in Sect. 3.1).

The size histogram bins utilised by the microcontroller are essentially levels of peak amplitude light signals scattered from measured particles. The bin boundaries are interpreted into particle diameters based on calibration using both theoretical and experimental data.

## 2.4 Calibration

### 2.4.1 Theoretical instrument response

An OPC directly measures the scattering cross-section of a particle. The scattering cross-section is a function of the particle properties (size, shape, refractive index), the collection angle of the instrument, and the wavelength of the beam (Bohren and Huffman, 1998; Rosenberg et al., 2012). This is defined as:

$$\sigma_{sca} = \frac{1}{k^2} \int\limits_{0}^{2\pi} \int\limits_{0}^{\pi} \frac{1}{2} \left[ |S_1(\theta, kD_p, n)|^2 + |S_2(\theta, kD_p, n)|^2 \right] \sin\theta \, \omega(\theta, \phi) d\theta d\phi \tag{1}$$

where:

$\sigma_{sca}$ = scattering cross-section, m$^2$

$k$ = wavenumber in medium, m$^{-1}$

$S_1$ = amplitude scattering matrix for parallel polarised light

$S_2$ = amplitude scattering matrix representing perpendicularly polarised light

$\theta$ = scattering angle measured from the incident beam direction, rad

$\phi$ = angle of the scattered light azimuthally around the incident beam, rad

$D_p$ = diameter of the scattering particle, m

$n$ = refractive index of the particle

$\omega$ = weighting function based on the mirror geometry

The weighting function can have values between 0 and 1 and describes the azimuthal extent of the collection optics in a finite element $\theta \to d\theta$ (Rosenberg et al., 2012). Optical particle counters often use annular reflectors, such as those in the Cloud Droplet Probe (CDP) (Lance et al., 2010) and the Passive Cavity Aerosol Spectrometer Probe (PCASP) (Cai et al., 2013), both by Droplet Measurement Technologies. The weighting function for these instruments can have values of 0 and 1 only. The UCASS utilises a non-annular reflector in the form of an elliptical mirror centred on a scattering angle of $\theta = 60$ ° with a half angle of 43.8 °. The mirror also has a central hole with half angle 10.7 °. Using this geometry, the weighting function, $\omega(\theta, \phi)$, is given by:

$$\omega(\theta, \phi) = \frac{1}{\pi}(\phi_m(\theta) - \phi_h(\theta)) \tag{2}$$

where $\phi_m$ and $\phi_h$ express the angular extent of the mirror and the hole, respectively, azimuthally around the laser beam as a function of scattering angle, $\theta$. For a primary reflector not centred on $\theta = 0$ °, $\phi$ is given by:

$$\phi_{m,h} = \begin{cases} \cos^{-1}[(\cos H_{m,h} - \cos L_{m,h} \cos \theta)/(\sin L_{m,h} \sin \theta)] & \text{if } -1 < (\cos H_{m,h} - \cos L_{m,h} \cos \theta)/(\sin L_{m,h} \sin \theta) < 1 \\ 0 & \text{if } (\cos H_{m,h} - \cos L_{m,h} \cos \theta)/(\sin L_{m,h} \sin \theta) \geq 1 \\ \pi & \text{if } (\cos H_{m,h} - \cos L_{m,h} \cos \theta)/(\sin L_{m,h} \sin \theta) \leq -1 \end{cases} \tag{3}$$

where:

20  $L_{m,h}$ = the lens angle of the mirror and hole, respectively

$H_{m,h}$ = the half angle of the mirror and hole, respectively

Using this weighting function in Eq. 1, we obtain the scattering cross-sections as a function of diameter. The matrices $S1$ and $S2$ are computed using Jan Schäfer's mie scattering code 'Matscat' (Schäfer, 2011; Schäfer et al., 2012; Bohren and Huffman, 1998; Lee, 1990; Kerker, 1969). Figure 7 shows the scattering cross-sections for the materials used in calibration: PolyStyrene-Latex (PSL), borosilicate glass and soda-lime glass, and two typical materials measured with the UCASS: water and mineral dust. Typically for OPCs, the relationship between scattering cross-section and geometric diameter is highly non-monotonic due to the presence of mie oscillations. These mie oscillations are most prevalent in the forward scattering region, and so the

wide collection angle of the UCASS elliptical mirror results in a near-monotonic relationship, as evidenced in Fig. 7. For each calibration standard used, the sizing error associated with mie oscillations falls well within the manufacturer stated standard deviation of the particle size, and so uncertainties associated with mie oscillations are not further considered.

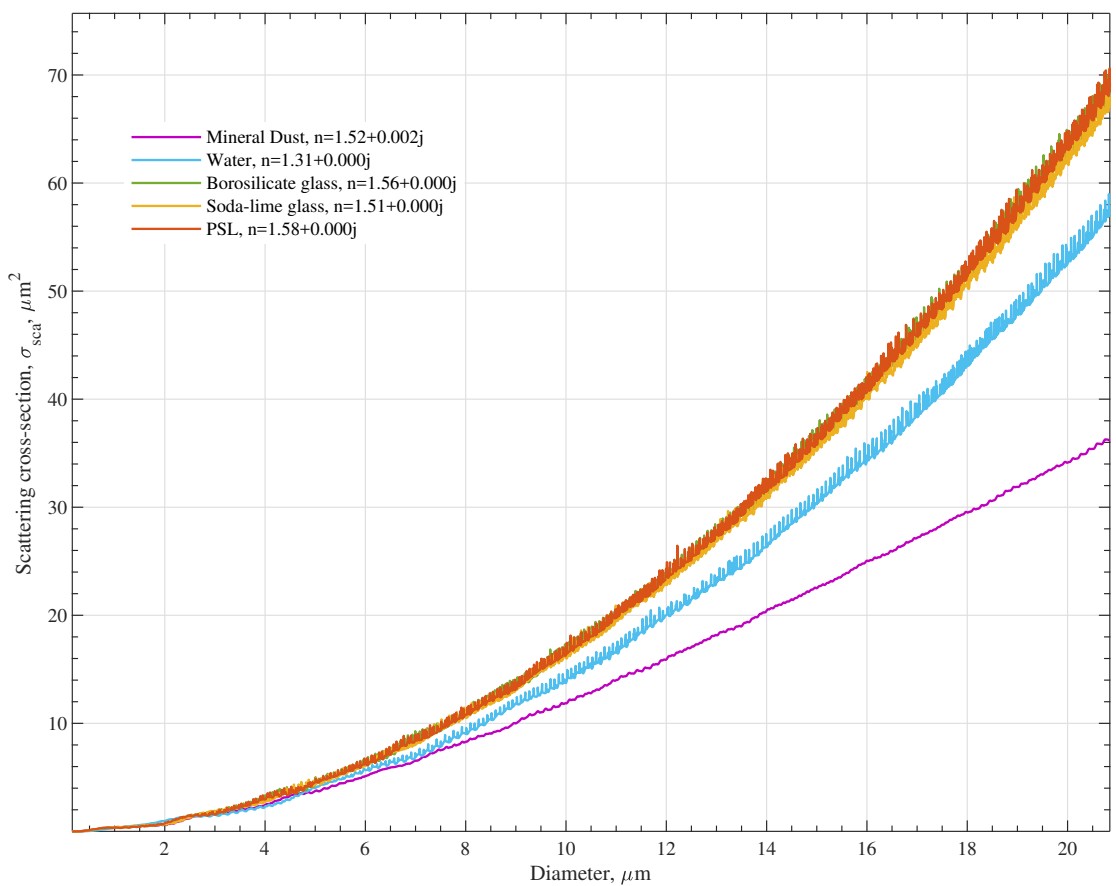

**Figure 7.** Scattering cross-sections, $\sigma_{sca}$, as a function of geometric diameter for mineral dust, water, borosilicate glass, soda-lime glass and PSL.

The scattering cross-section directly describes the amount of light collected by the mirror in the instrument. The instrument response is then a function of scattering cross-section, amplifier gain, detector sensitivity and laser power. Variations in the measured output between different units can occur due to manufacturer tolerances in detector sensitivity or laser power, thus causing offsets from the theoretical instrument response. Therefore, to constrain the calibration curve, we combine measured instrument responses with the theoretical results from mie theory.

### 2.4.2 Calibration Measurements

To constrain the calibration curve, we take measurements of the instrument response for a series of NIST-traceable particle standards. These include: PSL from Polysciences; borosilicate glass from Duke scientific; and soda-lime glass from Whitehouse Scientific. A drying column is used to ensure the proper drying and dispersion of the calibration standards, whilst also establishing a stable, non-turbulent air flow through the UCASS. The experimental set-up is shown in Fig. 8. The air intake consists of an initial drying chamber filled with silica gel desiccant, which dries the ambient air before passing it though a HEPA filter. This air is then directed through a series of pipes at the bottom of the drying column, which form a concentric ring around the aerosol input, thus establishing a clean, dry sheath flow for the aerosol to be injected into. The purpose of the sheath flow is to accommodate the large volume of air flowing through the instrument. By having a clean, dry sheath flow, calibration particles only need to be contained within the centre of the airflow, allowing them to travel through the sensing area, therefore fewer particles are wasted in the process. The main drying column consists of a 1 m tall tube with an inner tube for the airflow. The outer tube is packed with silica gel desiccant and the inner tube is lined with a microporous membrane. This combination aids the drying of any aerosol injected into the sheath flow will be fully dried before reaching the UCASS at the top of the column. A vacuum pump is used in line with the UCASS to establish an air flow of 3–5 ms$^{-1}$ (the intended operational velocity of the UCASS). Two different aerosol input techniques are used for wet and dry dispersion. The PSL microspheres are suspended in an aqueous solution and are aerosolised using a TSI Tri-Jet 3460 which is fed directly into the sheath flow. To disperse the dry calibration standards, a small amount of beads are placed into a small dispersion bottle consisting of an inlet and outlet pipe. A burst of dried, filtered air in the inlet pipe creates a turbulent environment inside the bottle which aids the breakup of any agglomerates. The outlet pipe is then directed into the sheath flow and the dispersed particles are carried in the airflow through the UCASS.

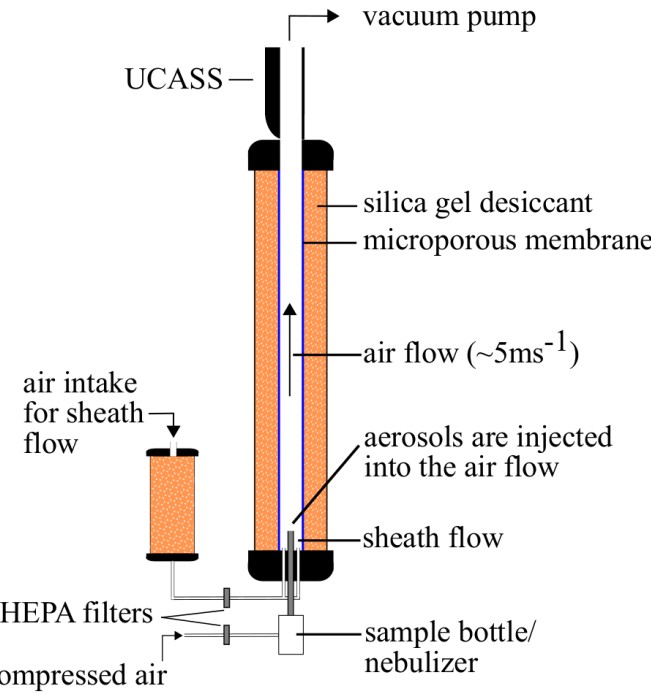

**Figure 8.** Experimental set up used to take calibration measurements. A vacuum pump is used to draw air through the drying column. The air intake is taken from ambient air which is first passed through desiccant and then a HEPA filter. This clean dry air is then directed through a concentric ring of pipes at the bottom of the drying column to create a clean, dry sheath flow. The aerosol is then injected into the sheath flow via the TSI tri-jet aerosol generator (for wet dispersion), or via a sample bottle for dry particles. The sheath flow constrains the aerosol in the centre of the air flow, which is carried directly through the sensing area of the UCASS.

To avoid sizing uncertainties associated with bin-width during calibration, a calibration-specific firmware is installed which allows Particle-by-Particle (PbP) pulse heights to be recorded. Measurements of various calibration standards are shown in Fig. 9. The top panel shows measurements from the 'high-gain' version of the UCASS, which has a nominal sizing range of 0.4–17 μm, and the bottom panel shows measurements from the 'low-gain' version of the UCASS which has a nominal sizing range of 1–40 μm. Although PSL and glass beads are typically considered monodisperse for calibration purposes, the full size range can be broad. Therefore the full size distribution of the aerosol is measured, and the mean, median and modal value of the whole distribution are found. Manufacturers may describe the average size of their test particles using different averages, and therefore it is necessary to select the correct average for each sample type. For these calibrations, the mean values are used for PSL and soda-lime glass, whereas the median is used for borosilicate glass. In cases where the full distribution is not captured, the modal value is used. This occurs in the case that the full size range of a particular sample may extend beyond the measurable limits of the instrument. For example, in Fig. 9, the largest sample used in this calibration is 14.4 μm, and it can be seen that the distribution is cut off at the point where the detector saturates. The same occurs for the low gain version for

the 37.36 μm sample. In these cases, the full distribution could not be measured, and therefore the mean or median cannot be used, and therefore the modal value is used instead.

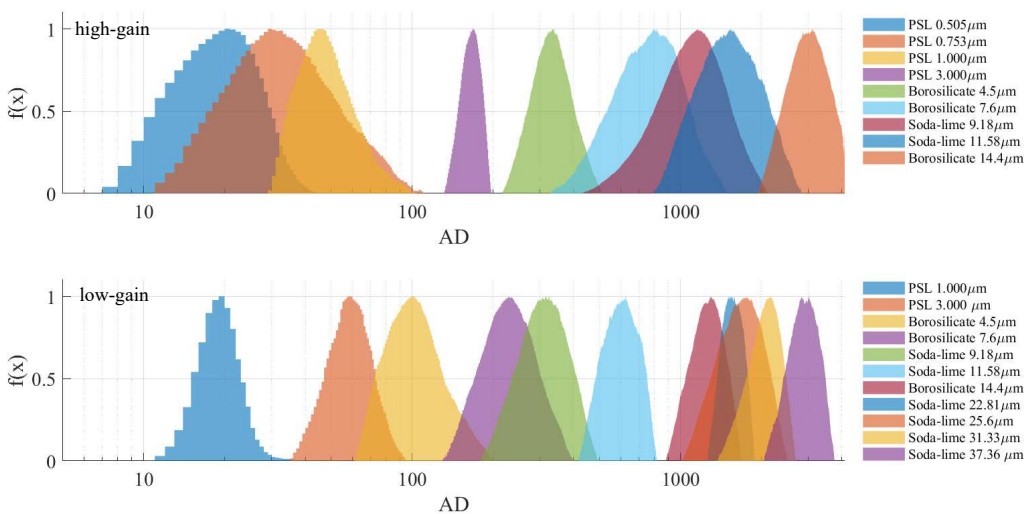

**Figure 9.** Measured instrument responses for various NIST-traceable calibration standards including PSL, Borosilicate glass, and soda-lime glass. $f(x)$ is the normalized counts as measured by the instrument, for presentation purposes, each distribution is normalised to a peak height of one. The top panel shows measurements conducted using a 'high-gain' version of the UCASS, and the bottom panel shows measurements conducted using a 'low-gain' version.

Theoretically, the instrument response is directly proportional to the scattering cross-section. However, the instrument response may differ to the theoretical response due to manufacturer tolerances in the optical and electronic components. Some variations such as laser power and amplifier gain will cause constant offsets over the measurement range. However, some offsets may be non-linear due to the varying Signal to Noise Ratio (SNR) across the measurement range and non-linearity in the detector sensitivity. To account for all offsets, the log of instrument response (Analogue to Digital Counts, $AD$) is plotted against the log of the scattering cross-section, and a line of best fit is applied as shown in Fig. 10.

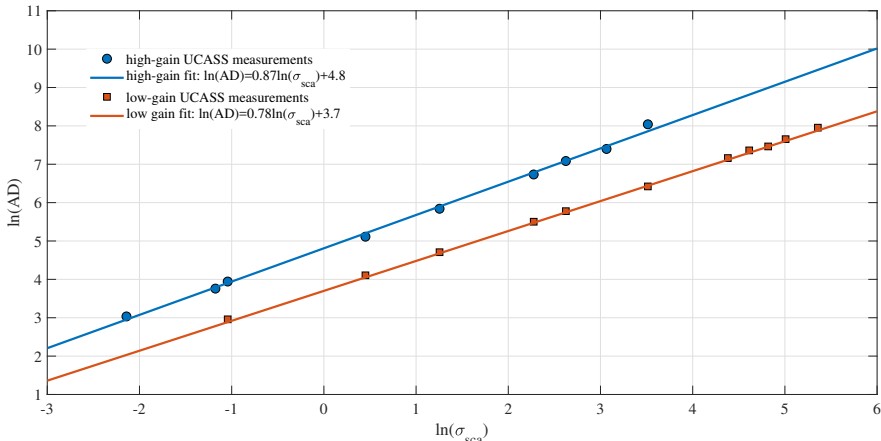

**Figure 10.** The log of the measured instrument response $(AD)$ is plotted against the log of the scattering cross-section $(\sigma_{sca})$ for the calibration standards used. For PSL and Duke Scientific borosilicate glass beads, the mean instrument response is used as this is how the manufacturer defines the particle size. For the Whitehouse Scientific soda-lime glass beads, the sample size is defined by the median and so the median instrument response is plotted.

A linear fit is applied to link the scattering cross-section to the instrument response. For the examples shown, the relationship for the high-gain UCASS is given by:

$$ln(AD) = 0.87ln(\sigma_{sca}) + 4.8 \tag{4}$$

and the relationship for the low-gain UCASS is given by:

$$ln(AD) = 0.78ln(\sigma_{sca}) + 3.7 \tag{5}$$

These equations can then be applied to any particulate (e.g. mineral dust, water) by computing the scattering cross-sections for discretized diameters across the measurement range and relating the instrument response to the particle diameter. Using this technique, finalised calibration curves are produced for five materials: PSL, soda-lime glass, borosilicate glass, water and mineral dust, as shown in Fig. 11. If a-priori knowledge of the aerosol type is known, then the calibration curve can be chosen

10  to best suit the application. The calibration curve can then be used to select the boundaries of up to 16 size bins (this is limited to 10 if interfacing the UCASS via XDATA protocol). Since the relationship between $ln(AD)$ and $ln(\sigma_{sca})$ remains valid for any material/shape of aerosol, if post-factum information on the particle shape or refractive index becomes available (i.e. via analysis of co-located measurements), then a post calibration can be applied.

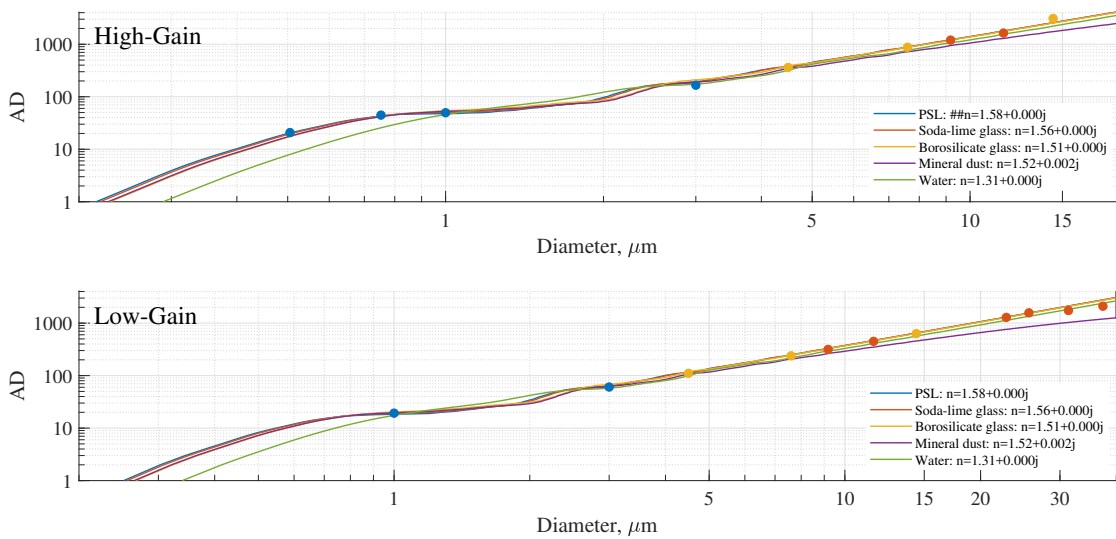

**Figure 11.** Final calibration curves for a high-gain (top) and low-gain (bottom) UCASS. The geometric diameter is plotted against instrument response for varying materials. The measured values (as shown in Fig. 9) are overlayed on the calibration curves and shown as filled circles, where the colour denotes the material. If a-priori knowledge of the refractive index is known, the calibration curve can be selected to best suit the material being measured. If post-factum information about the material becomes available, a post calibration can be applied.

## 2.5 Evaluation of Air Flow Uncertainty

The UCASS is a naturally-aspirated system, by which an external air flow (relative to the UCASS) is required to transport particles through the sensing area. The main advantage of this setup is the reduction of particle loss mechanisms induced by complex aerodynamic systems and non-isokinetic sampling, which has been evaluated empirically for artificially-aspirated systems (Spiegel et al., 2012; Hangal and Willeke, 1990; Von Der Weiden et al., 2009). However, the sampling efficiency is still dependent on the axial characteristics of the external airflow, hence a range of accepted (aerodynamic) operating conditions must be defined. To understand the airflow through the sample area, Computational Fluid Dynamics (CFD) using the 'Star CCM+' commercial code is used to model the air flow through the instrument at varying angles of attack. The CFD simulation is a 2-dimensional model on the symmetry plane of the UCASS, this was chosen because the velocity of an 'air parcel' in a subsequent plane to this is likely to be similar, meaning the viscous stress between planes is negligible. The fundamental equations behind this CFD are the Reynolds-Averaged Navier-Stokes (RANS) equations, which were chosen for computational efficiency when compared to direct numerical simulation.

Figure 12 shows the air flow through the UCASS for an external velocity of 5 $ms^{-1}$. For an axial air flow, the air velocity through the sample area is 5.6 $ms^{-1}$, 12 % higher than the external air flow. Whilst the drag on the inside walls of the UCASS cause the airflow to slow down, the sample area lies outside this boundary layer, and is therefore subject to a slightly higher air speed. The UCASS inlet is asymmetric due to the design constraints making it compatible with the KITsonde dropsonde

system and therefore, we must also consider the effect of tilt in different directions. As the UCASS is tilted upwards, the air flow through the sample area increases, however as it is tilted downwards, the air speed decreases. This relationship between tilt and air speed is asymmetric, with negative angles of attack having a greater impact on the air flow. Therefore, to retrieve reliable concentration measurements from the UCASS, the airflow or angle of attack must be known.

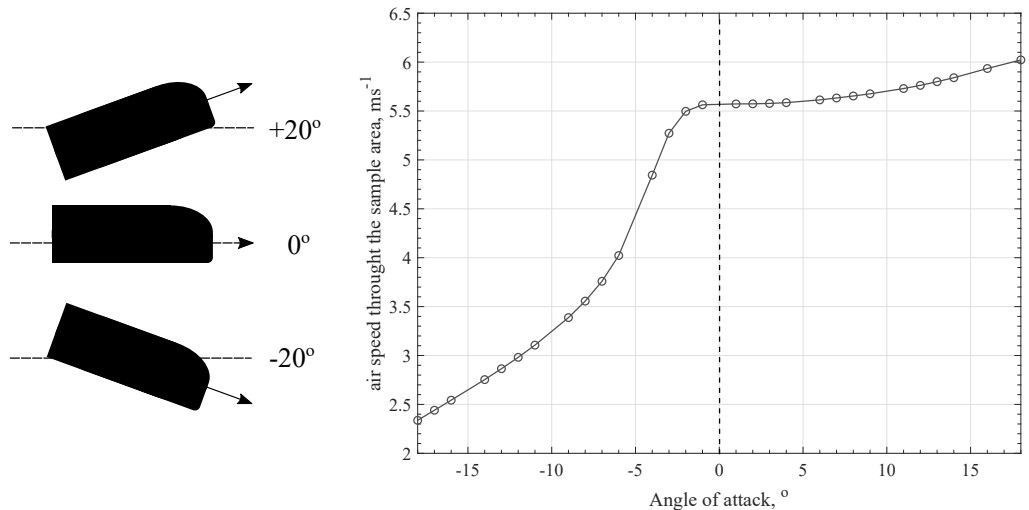

**Figure 12.** Model results showing the air velocity in the sensing area. The left panel shows how the tilt direction is described in relation to the instrument geometry. The right hand panel shows the simulated flow velocity in the sample for tilt angles between -20 ° and 20 ° in an external air flow of 5 ms$^{-1}$.

5    When the UCASS is launched on a meteorological balloon, the balloon-payload configuration acts as a pendulum system. To constrict the movement of the UCASS, the payload is configured as a double pendulum, whereby the UCASS is secured by a line below the balloon, and the meteorological sonde is secured below the UCASS. This double pendulum configuration allows fast energy dissipation, while at the same time ensuring small amplitude of the UCASS oscillations, generally smaller than the amplitude of the lowest element, the radiosonde. Although the UCASS still oscillates, model results show that the tilt

10  is constrained to within ±5 ° as shown in Fig. 13.

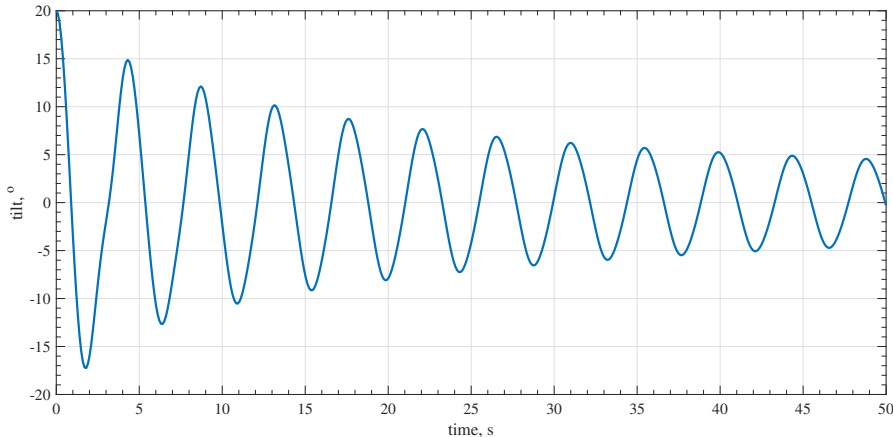

**Figure 13.** Model results showing the angular tilt of the UCASS with respect to the direction of ascent. Initial oscillations can reach $\pm\,20\,^{\circ}$ although these are damped after $\approx 30$ s, and constrained within $\pm\,5\,^{\circ}$.

By constraining the tilt of the UCASS within these limits, model results show that the mean air speed through the sample area is equal to $5.4\,\mathrm{ms}^{-1}$ with a standard deviation of $0.3\,\mathrm{ms}^{-1}$ which is used to compute the number concentrations. For dropsonde systems, a similar configuration is used, whereby the sonde is tethered between the parachute and the meteorological sonde, and the angle of tilt is constrained to within the same boundaries.

## 3   Results

### 3.1   Laboratory inter-comparisons

Inter-comparison tests were conducted in laboratory settings to assess the sizing and counting ability of the UCASS in comparison to reference instrumentation.

### 3.1.1   Sizing comparisons

To assess the sizing ability of the UCASS, a TSI 3300 Optical Particle Sizer (OPS) is used to measure the same particle calibration standards as discussed in Sect. 2.4.2. The TSI OPS is an optical particle counter, capable of sizing and counting particles in the 0.3–10 µm range in 16 user-configurable size bins. For comparative purposes, the TSI was set to the finest size resolution for each measurement, focusing only on the size range of interest. The results are compared to the measurements taken by the UCASS while in Particle-by-Particle mode (as discussed in Sect. 2.4.2). Measurements are also compared with statistical values given by the manufacturer of the particle standard, where available. These measurements are shown in Fig. 14.

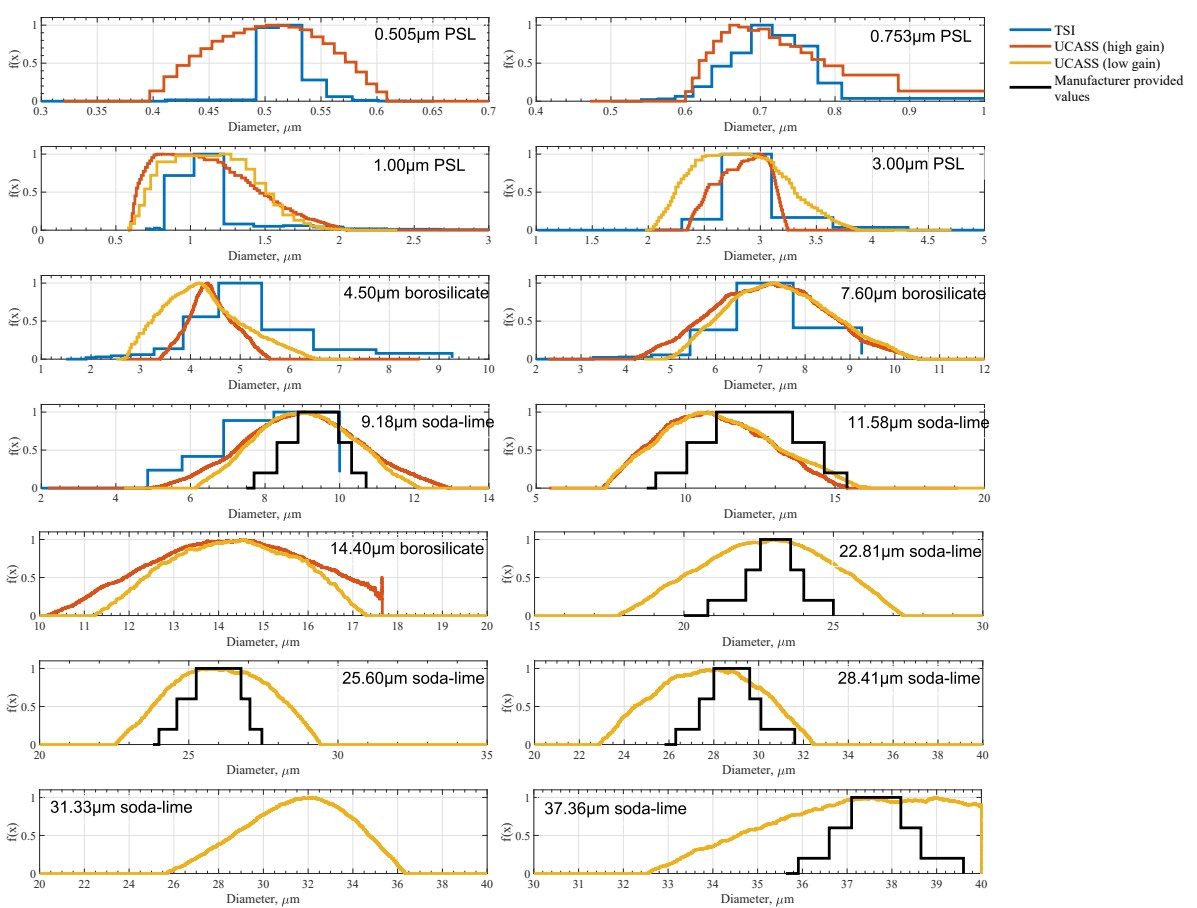

**Figure 14.** Measurements of particle size distribution from the TSI 3330 OPS, the high-gain UCASS and the low-gain UCASS. Manufacturer values are also shown, where available. Data for PSL and borosilicate standards are summarised in Table. 1 while data for soda-lime samples are summarised in Table. 2

.

| Manufacturer values (mean and sd), μm | Material | TSI | UCASS (High Gain) | UCASS (Low Gain) |
|---|---|---|---|---|
| $0.505 \pm 0.016$ | PSL | $0.507 \pm 0.02$ | $0.500 \pm 0.05$ | - |
| $0.753 \pm 0.01$ | PSL | $0.697 \pm 0.07$ | $0.696 \pm 0.07$ | - |
| $1 \pm 0.01$ | PSL | $1.029 \pm 0.34$ | $1.114 \pm 0.35$ | $1.117 \pm 0.4$ |
| $3 \pm 0.07$ | PSL | $2.700 \pm 0.25$ | $2.903 \pm 0.22$ | $2.823 \pm 0.44$ |
| $4.5 \pm 0.4$ | borosilicate | $4.560 \pm 1.13$ | $4.419 \pm 0.39$ | $4.373 \pm 0.74$ |
| $7.6 \pm 0.4$ | borosilicate | $6.493 \pm 1.05$* | $7.35 \pm 1.20$ | $7.53 \pm 1.12$ |
| $14.4 \pm 0.8$ | borosilicate | - | $14.418 \pm 1.66$ | $14.394 \pm 1.31$ |

**Table 1.** Numerical summary of data shown in Fig. 14 pertaining to PSL and borosilicate particle standards. The first column shows the mean and standard deviation of calibration standards as given by the manufacturer. Remaining columns show distributions measured by the TSI OPC, the high gain UCASS and low gain UCASS. Incomplete distributions (i.e. where the full size distribution exceeds beyond the measurable size range) are denoted with an asterisk.

| Manufacturer values (median & IQR) | TSI | UCASS (High Gain) | UCASS (Low Gain) |
|---|---|---|---|
| $9.18$ ($iqr = 1.2$) | $6.889$ ($iqr = 1.4$)* | $9.17$ ($iqr = 2.14$) | $9.14$ ($iqr = 1.8$) |
| $11.58$ ($iqr = 2.4$) | - | $11.07$ ($iqr = 2.5$) | $11.18$ ($iqr = 2.7$) |
| $22.81$ ($iqr = 1.1$) | - | - | $22.88$ ($iqr = 3.0$) |
| $25.60$ ($iqr = 1.4$) | - | - | $26.09$ ($iqr = 2.4$) |
| $28.41$ ($iqr = 1.7$) | - | - | $27.86$ ($iqr = 3.2$) |
| $31.33$ | - | - | $31.72$ ($iqr = 3.2$) |
| $37.36$ ($iqr = 1.1$) | - | - | $38.38$ ($iqr = 3.5$)* |

**Table 2.** Numerical summary of data shown in Fig. 14 pertaining to soda-lime glass particle standards. The first column shows the median and interquartile range ($iqr$) of calibration standards as given by the manufacturer. Remaining columns show distributions measured by the TSI OPC, the high gain UCASS and low gain UCASS. Incomplete distributions (i.e. where the full size distribution exceeds beyond the measurable size range) are denoted with an asterisk.

### 3.1.2 Counting comparisons

To assess the counting ability of the UCASS, an aerosol chamber was filled with polydisperse aerosol. The polysdisperse aerosol was generated using vapourizer and a mixture of propylene-glycol and vegetable glycerine. The TSI was mounted in the chamber and sampled continuously, the UCASS was mounted nearby the TSI, and a pump attachment was used to draw air through the inlet. Measurements were repeated using different size bins in order to collect data across the full measurable range (0.4–10 μm), and with a higher resolution at smaller sizes. Typical concentrations during the experiment were of the order $10^2$ cm$^{-2}$. The measured concentrations of the UCASS are compared to the measured concentrations of the TSI in Fig. 15.

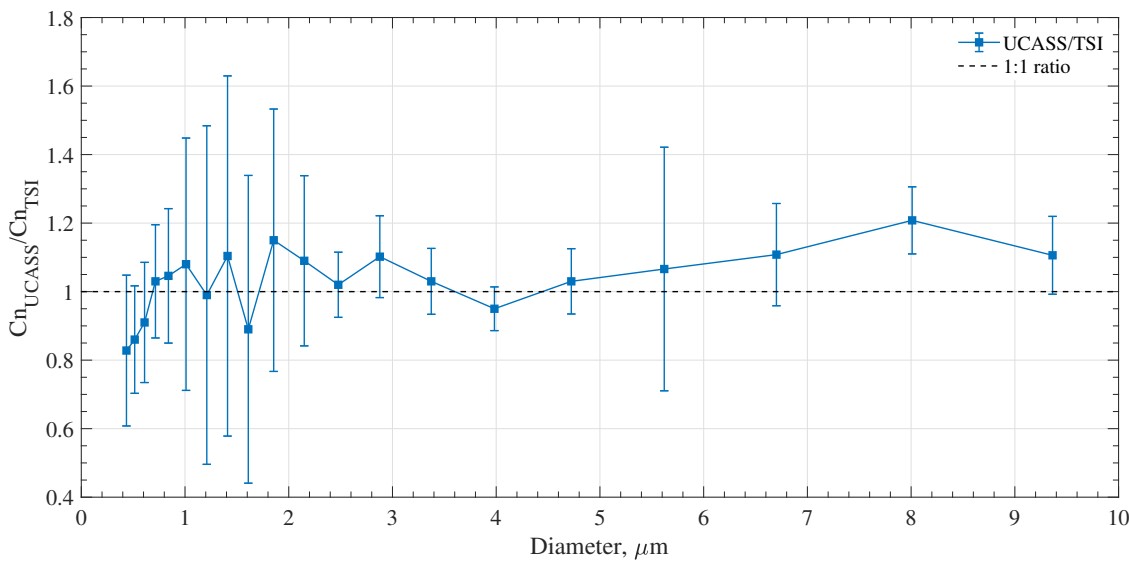

**Figure 15.** Ratio of UCASS measured number concentrations ($Cn_{UCASS}$) to TSI measured number concentrations ($Cn_{TSI}$ where data points represent means averaged over 100 readings, and error bars represent standard deviations. The 1:1 ratio is shown as a black dashed line.

### 3.2 In-situ inter-comparisons

A low-gain configuration of the UCASS (calibrated for water droplets) was tested at the Observatoire de Physique du Globe de Clermont-ferrand (OPGC) observatory, located atop the Puy De Dôme volcano at an altitude of 1465 m. The observatory consists of a rooftop measurement platform and an inbuilt wind tunnel capable of pumping through the passing clouds. As the UCASS requires an external air flow, the UCASS was tested in the wind tunnel, co-located with a Cloud Droplet Probe (CDP-2) from Droplet Measurement Technologies (DMT) (Lance et al., 2010). The CDP is an open geometry system which provides size and concentration measurements in the range 2–50 μm, and is intended for air speeds of 10–250 ms$^{-1}$. During a stratocumulus event, the UCASS and CDP were co-located in the wind tunnel, and the cloud was drawn through at varying speeds from 5 ms$^{-1}$ (the lowest operational velocity of the wind tunnel) to 20 ms$^{-1}$. Figure 16 shows the measured Liquid Water Content (LWC) by the UCASS (blue) and the CDP (orange). Both instruments measured at a rate of 1 Hz (shown by dotted lines), and the solid lines represent the smoothed data, using a 10-point moving average. The air speed is shown via a green dotted line corresponding to the right-hand y axis. It can be seen that, throughout the experiment, both the UCASS and CDP capture the same temporal variations in the liquid water content, although the magnitudes start to diverge at air velocities above 17 ms$^{-1}$. This under-counting by the UCASS is expected due to the short ToF (discussed in Sect. 2.3 which causes the particles to be rejected. As discussed in Sect. 2.3, this ToF threshold is intended to prevent miscounting of short pulses which can be caused by electrical noise. However, the ToF limits can be altered to suit the measurement platform. In the overlapping measurement range (air speeds between 10 and 15 ms$^{-1}$), the ratio of UCASS measured LWC to CDP measured LWC is equal to 1.02.

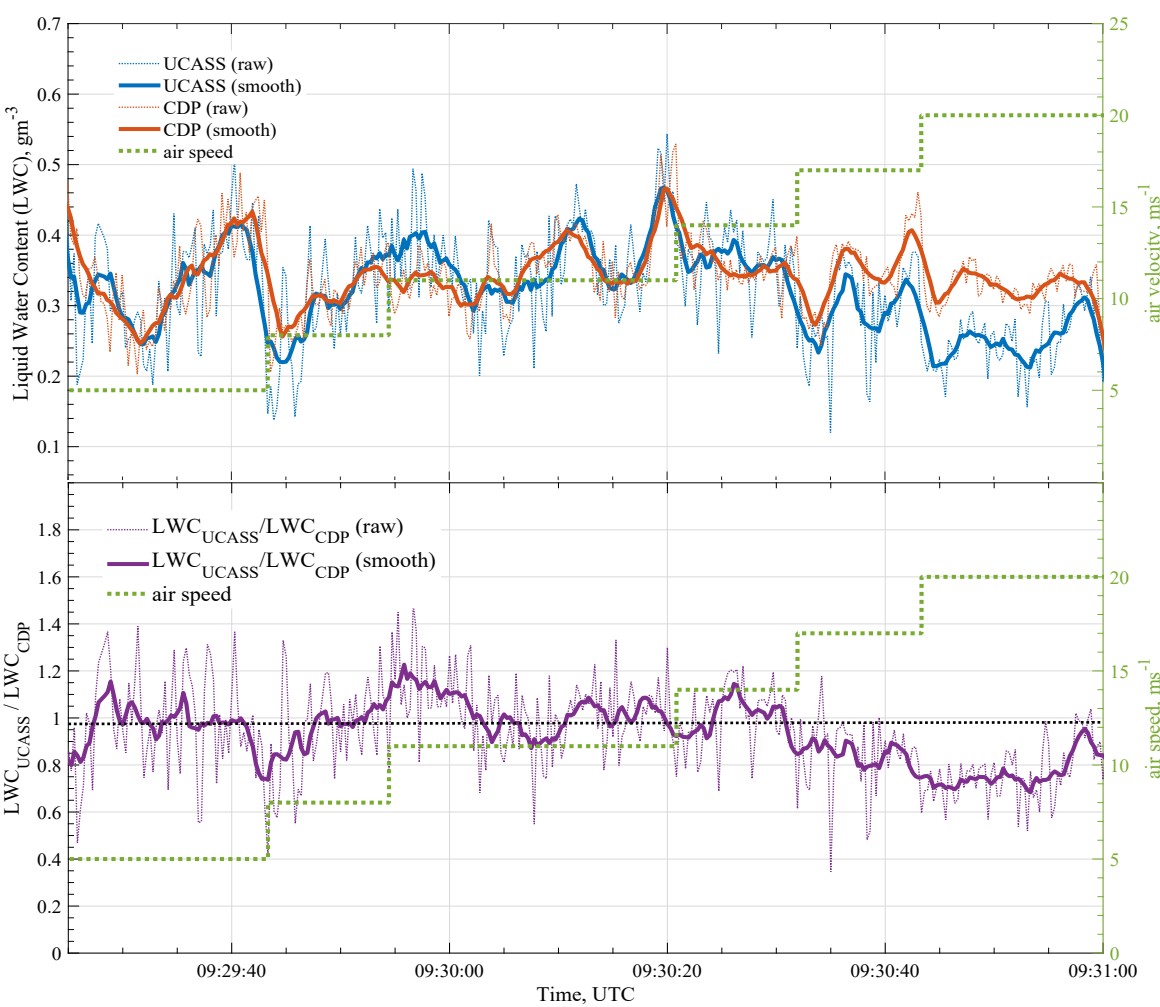

**Figure 16.** Top: measured Liquid Water Content (LWC) by co-located UCASS and CDP instruments at varying air speeds. The two instruments were installed side-by-side in the wind tunnel at Puy de Döme during a stratocumulus event and the air speed was increased from 5 ms$^{-1}$ to 20 ms$^{-1}$. Bottom: Ratio of the LWC measured by the UCASS to the LWC measured by the CDP. The black dotted line shows the 1:1 ratio.

The data in the following section was collected prior to the availability of the borosilicate glass standards. As such, the instruments usedwere calibrated using a 5+ point calibration with PSL and soda-lime glass.

### 3.2.1 Dropsonde system

Dropsondes were launched from the Dornier 128 aircraft north of Magdeburg, Germany, on the 3rd of August 2013. The
5   packages contained the UCASS in tandem with KITsondes (Wieser et al., 2014), for co-located particle and meteorological data. The size distribution profiles measured by the UCASS indicated mineral dust present in the free troposphere during the 3rd

August for a drop conducted at 1310UTC. At 1534UTC, AERONET sun photometry retrievals (Dubovik and King, 2000) from IfT Leipzig (located 100 km South-South-East of Magdeburg) confirmed the presence of dust. The integrated size distribution measured by the UCASS is compared to the AERONET inversion from Leipzig in Fig. 17(a). The size distribution is integrated for the column between 3 and 5 km, to exclude boundary layer aerosol over the Colbitz-Letzlinger Heide area (the EDR74

exclusion zone). Figure 17(b) shows the dust dispersion model for 1200UTC using the MACC (Monitoring Atmospheric Composition and Climate) forecast, showing the transport of dust over Germany from the Sahara. The locations of Magdeburg and Leipzig are marked by red and yellow crosses, respectively.

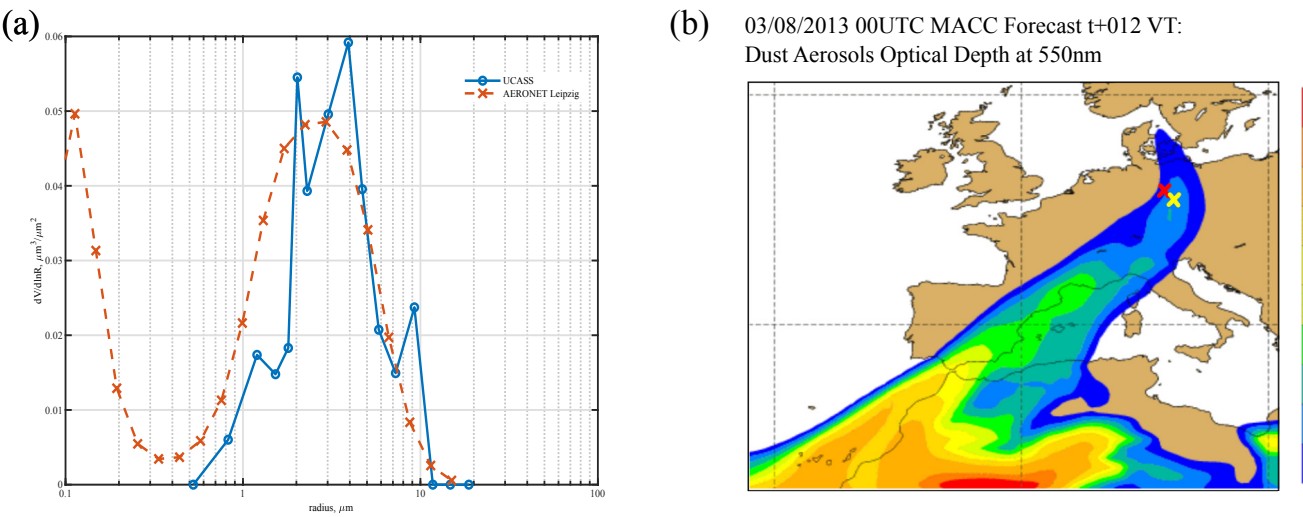

**Figure 17.** (a) Integrated aerosol size distribution measured by the UCASS compared with the AERONET inversion (as retrieved from the sun photometer data) from Leipzig. (b) Dust dispersion model showing Saharan dust transport over Leipzig (marked by a yellow cross) and Magdeburg (marked by a red cross).

In a further drop during this dust event, a UCASS dropsonde system was depoyled 3 minutes later at 1313UTC. This payload measured a thin, embedded cloud layer as shown by high number concentrations at larger sizes at 4 km. This is illustrated in

Fig. 18(a), where the number concentrations for each of the 16 size bins are shown. Over the 6 km profile, the majority of particulates counted measured less than 3 μm in diameter, except for a thin layer at 4 km with high concentrations of particles in larger size bins. This change in concentration and size suggested the presence of a thin cloud layer, which was confirmed by the humidity profile measured by the attached KITsonde shown in Fig. 18(b), which measured high humidities at 4 km. Figure 18(c) shows the effective diameter with respect to altitude, it can be seen that the saharan dust layers above/below the

cloud have an effective diameter of $\approx 2$ μm. At 4 km when the UCASS enters the cloud layer, this effective diameter steeply increases to $\approx 6$ μm. As discussed in Sect. 1, dust can cause the formation of clouds by acting as a cloud condensation nuclei, and so embedded clouds during dust events are not uncommon. However, the presence of cloud presents difficulties for remote sensing instrumentation, and therefore retrievals are not possible for cloud-contaminated datasets. So long as the UCASS is

launched in tandem with a meteorological sonde, the measured humidity profiles can be used to differentiate between cloudy and non-cloudy conditions, and therefore the UCASS can be used to measure both cloud and aerosol size distributions even when both are present.

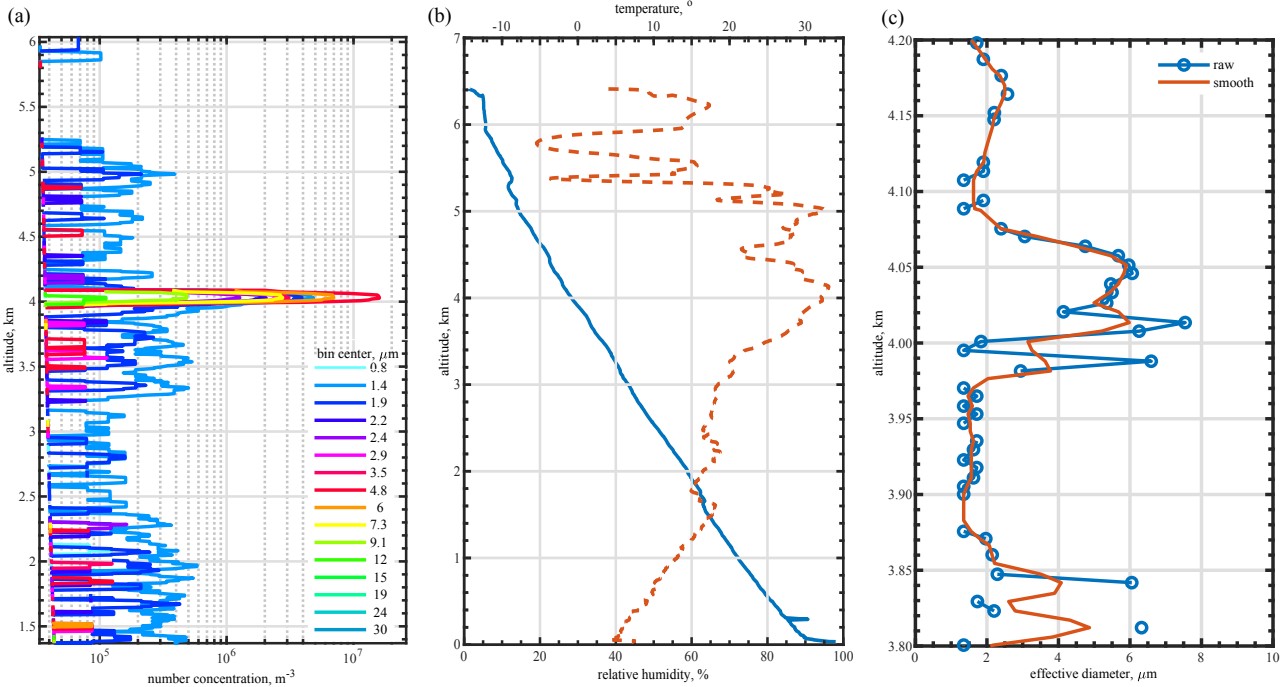

**Figure 18.** (a) Number concentration of particulates in the 16 UCASS size bins with respect to altitude, suggesting an embedded cloud layer at 4 km. (b) Temperature and relative humidity profiles from the attached KITsonde, also suggesting an embedded cloud layer at 4 km with a spike in humidity. (c) Effective diameter with respect to altitude, the embedded cloud layer at 4 km is evident due to the steep increase in effective diameter. The blue trace shows raw data, whilst the orange trace shows smoothed data using a 10 point moving average.

### 3.2.2 Upsonde system

5   The UCASS was used during the AERosol properties – Dust (AER-D) and Sunphotometer Airborne Validation EXperiment - Dust (SAVEX-D) campaigns over Cape Verde during August 2015 (Marenco et al., 2018). A UCASS was launched as part of a balloon-based system in tandem with a GRAW DFM-09 during a Saharan dust event on the 25[th] August, from Instituto Nacional De Meteorologia E Geofisica (INMG), Espargos, Sal island. The launch was conducted at 1701UTC from ground level (70 m above sea level), this launch was timed to coincide with a deep profile (profile 6) conducted by SAVEX-D flight

10   number B934. The research aircraft started the profile from an altitude of 4 km at 1710UTC off the West coast of Boa Vista island (≈ 0.75 °S, 0.15 °W of the balloon launch site). The aircraft then flew due North, passing Sal island until reaching an altitude of 690 m. Figure 19(a) shows the flight paths of the aircraft and balloon (up to 4 km), whilst Fig. 19(b) shows the altitude of both profiles with respect to time. Figure 19(c) shows the near real time dust forecast from the Met Office

Global Atmosphere model (Martin et al., 2018) for 1800UTC, where it can be seen that dust is advected westwards from the Sahara. The circular section shows a zoomed in image over the Cape Verde islands, and the islands of Sal and Boa Vista are highlighted by a red rectangle. It can be see from the forecast that the aircraft and balloon profiles were conducted at the leading edge of a dust layer and therefore, even though the spatial and temporal differences between the two profiles are minimal, some variability in the dust layer may be expected.

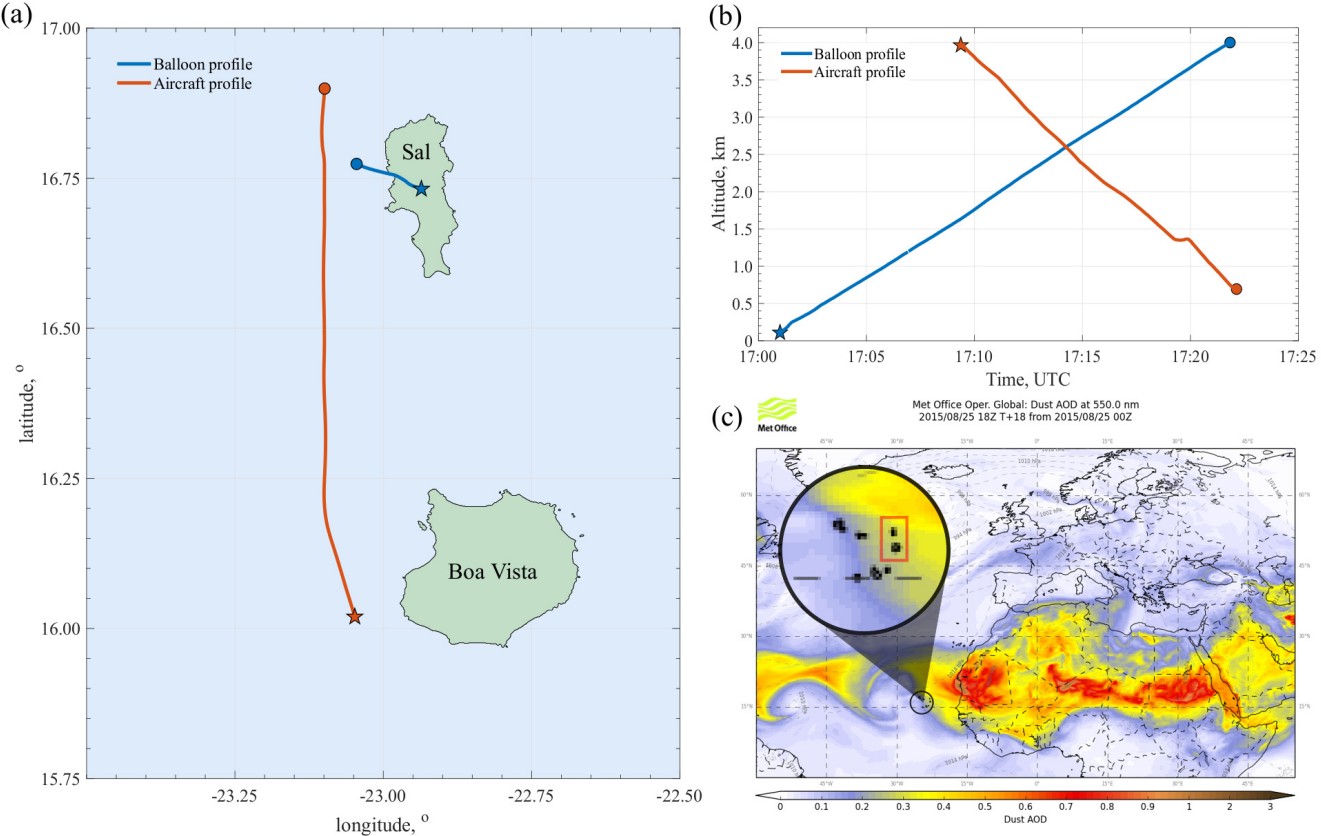

**Figure 19.** A balloon launch was conducted to coincide with profile 6 of SAVEX-D flight B934. (a) shows the flights paths of the balloon (blue) and aircraft (orange) during this profile, where the start and end points are marked by stars and circles, respectively. The altitudes are shown in (b), whereby the aircraft descends from an altitude of 4 km to 690 m. For comparative purposes, only the balloon data up to an altitude of 4 km are shown here. (c) shows the MET office global atmosphere model Aerosol Optical Depths (AODs) for 1800UTC. The cape verde islands are zoomed in and the islands of Sal and Boa Vista are highlighted in a red rectangle. It can be seen from this that the measurements were located to the edge of a dust layer, with anticipated AOD of $\approx 0.3$.

Figure 20(a) shows the mass concentration profile measured by the UCASS, compared with measurements from the aircraft mounted PCASP (Passive Cavity Aerosol Spectrometer Probe). The PCASP is an aircraft mounted OPC by Droplet Measurement Technologies. Similarly to the UCASS, the PCASP utilises wide-angle scattered light, with the primary reflector

covering the angular range 35–120 °. For comparative purposes, the mass concentrations shown only apply to the overlapping measurable size range of the UCASS and PCASP, in this case the size range is from 0.4–3 µm, corresponding to the first 7 bins of the UCASS. Figure 20(b) shows the ratio of the mass concentrations measured by the UCASS and PCASP, respectively. Over the 4 km profile, the average ratio is 1.2, with the greatest differences between the two measurements occur at the base

5    and top of the dust layer ($\approx$ 1.6 and 4 km respectively). The dust layer was advected from the East as shown in Fig. 19, thus the spatial separation between the two profiles explains the discrepancies. The more easterly UCASS measurement captures higher concentrations and a higher layer height, whilst the more westerly PCASP measurement measures a lower layer height, likely due to gravitational settling.

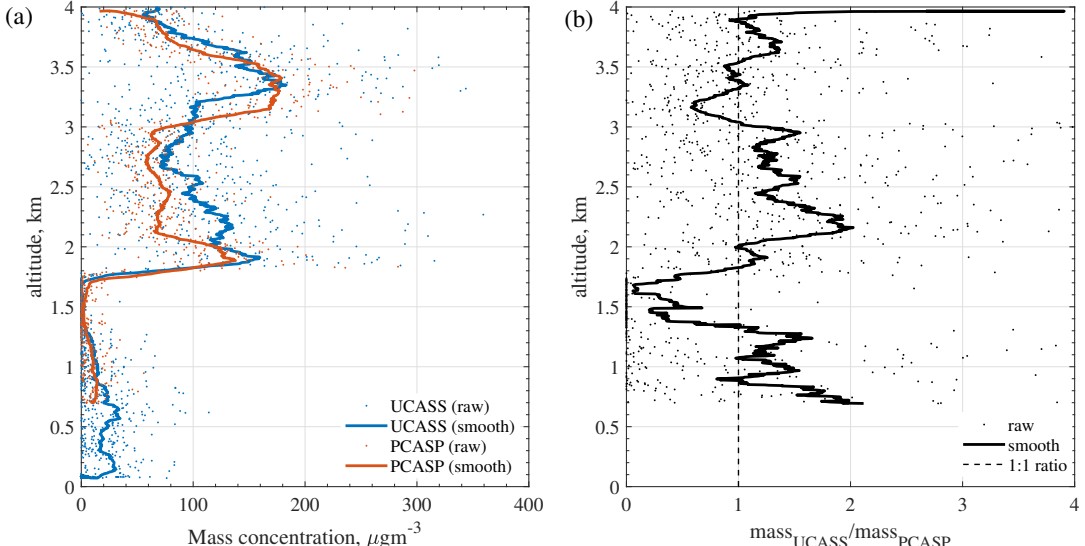

**Figure 20.** (a) shows the measured mass concentrations with respect to altitude for the balloon-borne UCASS (blue) and aircraft mounted PCASP (orange). The raw data for each instrument is shown via individual data points, and the smoothed data is shown by a solid line, using a 10 point moving average). The mass concentrations shown pertain to particles within the 0.4–3 µm size range for comparative purposes. (b) shows the ratio of the measured UCASS mass concentration to the PCASP measured mass concentration. It can be see that the largest differences occur at the top and bottom of the dust layer, which may be due to spatial variation and/or gravitational settling.

## 4   Conclusions

10   The Universal Cloud and Aerosol Sounding System (UCASS) is a lightweight, open-path Optical Particle Counter (OPC), designed for the measurement of micron-scale particles. The open-geometry design bypasses issues associated with narrow inlets such as clogging and counting uncertainties associated with complex air-flow systems. Furthermore, by removing the need for heavy or expensive pumps, the size, weight and cost of the instrument are kept to a minimum. The UCASS measures 180 mm long, 64 mm in diameter and weighs 280 g. The UCASS can then be used as a stand alone instrument whereby

data is logged autonomously via an on board SD card, or the UCASS can be interfaced via XDATA protocol with several commercially available meteorological sondes. Therefore, the UCASS is suitable to create low-weight payloads for dropsonde systems, balloon-borne systems or Unmanned Aerial Vehicles (UAVs).

The UCASS can be configured as a high-gain or low-gain mode, giving nominal measurable ranges of $\approx$ 0.4–17 µm and $\approx$ 1–40 µm, respectively, with up to 16 configurable size bins. As the UCASS measures predominantly side-scattered light, the sizing ability is not hindered by the presence of mie oscillations that can cause uncertainties with forward scattering probes. However, as side-scattered light is more dependent upon the refractive index of the scattering particle, the exact detection limits are dependent upon the refractive index of the material being measured. To ensure accurate sizing, an 8+ point calibration is applied using a variety of particle standards including PSL, soda-lime glass and borosilicate glass. To account for differences in refractive index, the geometric size is converted to scattering cross-section using mie-theory (Rosenberg et al., 2012), and the measured instrument response is plotted against scattering cross-section. For each unit, a relationship is found between the scattering cross-section ($\sigma_{sca}$) and instrument response ($AD$), in the form $ln(AD) = aln(\sigma_{sca}) + b$, where $a$ and $b$ are constants found through calibration measurements. By using scattering cross section, rather than geometric diameter, this equation can be applied to particles of any refractive index or shape. Typically, the bin boundaries are chosen using a-priori information on the particles being measured (i.e. mineral dust or water), however with this relationship established, post calibrations can be applied easily if post factum information of the aerosol properties becomes available. Due to this, the UCASS can be used to measure in mixtures of cloud and aerosol (as discussed in Sect. 3.2.1) where humidity profiles can be used to determine dry and cloudy air masses. The bin boundaries for different layers can then be modified for water or mineral dust accordingly.

The UCASS has been tested in a number of field and lab-based studies for the measurement of both aerosol and cloud droplets. During a Saharan dust in Magdeburg, Germany, the UCASS was used along the KITsonde as a dropsonde system, and dropped from the Dornier 128 research aircraft. The volume size distribution measured by the UCASS shows good agreement with the AERONET inversion conducted at Leipzig, 100 km South-South-East of Magdeburg. The UCASS has also been tested as an upsonde system during the AER-D/SAVEX-D campaign, conducted over Cape Verde in 2015. The UCASS was interfaced with a GRAW DFM-09 and launched from Sal island, coincident with a SAVEX-D research flight conducted 80 km South-South-West of the launch site. Comparisons of the mass concentration measurements from the UCASS and the aircraft-mounted PCASP show both instruments agreement within 20%. The largest discrepancies are associated with the top and bottom of the dust layer, as anticipated from model results which show the profiles to conducted on the edge of a dust layer. For more controlled tests, the UCASS was tested in a wind tunnel at Observatoire de Physique du Globe de Clermont-ferrand (OPGC) observatory, Puy De Dôme, during a stratocumulus event. The UCASS was co-located with a CDP-2 with a spatial separation of <0.5 m. The two instruments are designed for measurement in different ambient air speeds, but in the overlapping range (10–15 ms⁻¹), the liquid water content measurements from the UCASS and CDP agree within 2%.

*Code and data availability.* The AERONET data used in Fig. 17 is available online via the AERONET site: https://aeronet.gsfc.nasa.gov/. PCASP data used in Fig. 20 are taken from SAVEX-D flight B934, access to this dataset can be requested through the Centre for Environ-

mental Data Analysis (CEDA) and is archived at : CEDA/Data Server/badc/ice-d/data/bae-146/b934-2015-aug-25. CDP data used in Fig. 16 was provided by OPGC and is not publicly available. All UCASS data, plot data and analysis code can be made available upon request by contacting the lead author.

*Author contributions.* The original draft of the paper was prepared by HS and reviewed and edited by all co-authors. Funding was acquired

5 by ZU (NERC capital grant) and HS (Aerosol Society small research grant, Trans National Access grant). The project was conceptualized by ZU, PK, EH, WS, RK and HS. Investigations were conducted by HS, ZU, AW, MK and JG. Formal analysis and data curation were performed by HS, ZU and JG. HS, ZU, PK, CS, RK, WS, MK, RG and JG contributed to the methodology. Software was provided by WS.

*Competing interests.* The authors declare that they have no conflict of interest.

*Acknowledgements.* This work was funded by grants from the Natural Environment Research Council (NERC), numbers: NE/L002809/1

10 and CC0030, alongside a small research grant from the Aerosol Society. Experiments at Puy De Dome observatory were supported by a Trans National Access (TNA) grant through the ACTRIS-2 initiative, and CDP data were provided via staff at OPGC and TROPOS. The AERONET inversions used in Fig. 17 were provided by AERONET Leipzig. Met Office provided near real time forecast imagery for the AER-D/SAVEX-D campaign as used in Fig. 20:. Airborne data from AER-D/SAVEX-D were obtained using the BAe-146-301 Atmospheric Research Aircraft operated by Directflight Ltd and managed by the FAAM, which is a joint entity of NERC and the Met Office.

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
