# Peer review of "The Universal Cloud and Aerosol Sounding System (UCASS): a low-cost miniature optical particle counter for use in dropsonde or balloon-borne sounding systems."

_Atmospheric Measurement Techniques, 2019_

## Referee Comment (RC1) · Masatomo Fujiwara (Referee) · 31 Mar 2019

Review report

"The Universal Cloud and Aerosol Sounding System (UCASS): a low-cost miniature optical particle counter for use in dropsonde or balloon-borne sounding systems."

By H. R. Smith et al.

This paper describes a recently developed small-size optical particle counter for aerosol and cloud particle measurements for balloon and dropsonde applications and shows some field results in comparison with those from other particle instruments. The description of the instrument and calibration processes is basically comprehensive (though it seems there is a room for improvement), and the field results are very interesting and promising. I think the manuscript will be acceptable for publication in Atmospheric Measurement Techniques after considering the comments and suggestions below.

Major comments.

1. The instrument uses 658 nm laser light with an open-path configuration. Isn't there any influence of stray sunlight on the measurements? If not, is there any special measure for the daytime use?

2. The authors wrote at page 9 that Time-of-Flight (ToF) data is used to "reject" signals that may come from "a large body or agglomeration of particles." Does this mean that such particle signals are not on the record? There is alternative approach in that such signals (including ToF data) are also recorded, processed, but may be removed in the data analysis phase. Is there any reason to reject those at the onboard circuit? I ask this question because this treatment may miss cloud signals if particles are greater than 40 μm. For example, in Figure 14, at 5 km, there is another ~100 % relative humidity layer. It is not clear whether there was no cloud or there were clouds with particles much greater than 40 μm.

3. The description of the assembly, optical set-up, and sensing area is sometimes not easy to follow. I understand this because the instrument is three dimensional, but I have some ideas to improve this. I suggest the authors to (1) define the common x, y, and z axes for Figures 1 through 4, (2) show the axes explicitly in each of Figures 1 through 4, (3) avoid terminologies such as "width/depth", "left/right", "above/below" etc., but use the axis to specify the direction. Furthermore, please consider to use a common set of identifiers e.g., (a), (b), … in these four figures; for example, in Figure 2, (a)➔(b1), (b)➔(b2), (c)➔(b3), (d)➔(b4) (and keep Figure 1 as it is), and use e.g., (i), (ii), … instead of (a), (b), … in Figures 3 and 4. The mirror schematic in Figure 2 is very different from those in Figures 3 and 4, which made me confused at first. The

laser light schematic in red in Figures 2, 3, and 4 may be improved by making them more consistent across these figures, and add an explanation in caption Figure 5 about how the "major axis" and "minor axis" correspond to the laser light schematic in Figures 2 through 4. Finally, please add the dimension information to Figures 1 through 3 as much as possible, so that it becomes easier to read the text by referring to these figures.

Other comments.

Section 1 Introduction:
- I think MODIS and MISR are not lidars.
- There have been several particle instruments for balloon sounding. Some examples can be found in the Introduction of Fujiwara et al. (2016). Other instruments include LOAC (Renard et al., 2016) and POPS (Gao et al., 2016).
- I think that an OPC for dropsonde may be new. I assume that the dimension, shape, and configuration of the UCASS was determined so that it can be mounted in the dropsonde launcher. If so, please explicitly write the conditions under which an instrument can be used as a dropsonde. Also, the strength of the dropsonde over the balloon sonde system may be discussed (e.g., the former can be more easily targeted to a specific airmass including a specific cloud system).
- Because the examples of aerosol particle measurements shown in this paper are actually only for Saharan mineral dust, the role of mineral dust particles on the climate may be explicitly discussed.
- The mass of the instrument 280 g should be explicitly written here (near the term "lightweight"; and also in Section 2).

Section 2 Instrument Design, subsections 2.1 through 2.2:
- See the major comments.
- Why is the sensing area ("0.5 mm$^2$") an area, not a volume?
- It is helpful for the readers to summarize (e.g., to prepare a summary table for) the particle signal data (i.e., $I_1$, $I_2$, ToF, pulse height, etc.) and their relation with the criteria for data quality assurance (i.e., particle path inside/outside the sensing area, agglomeration of particles, etc.) Please also add the information how the numbers 0.4, 17, 1, and 40 μm for the particle size limits were actually determined.
- Figures 3 and 4 indicate that the detector only collect light reflected on the mirror. There is no contribution from the directly scattered light (i.e., around 120 deg.)?

Section 2.3 Electronics: Perhaps, the explanation of the "gain" i.e., "high gain version" and "low gain version" is first described here at page 8. In the current manuscript, the gain is mentioned first in the

Calibration section at page 13, and it is not very clear what the "gain" actually is.

Section 2.4.2 Calibration Measurements
- Have you used all the calibration particles for each of the two different gain versions? In Figure 8, the set of particles shown is different between the two. For example, what is the response of the high-gain version to soda-lime 37.36 μm particles?
- What were the ToF values for these experiments? How about the frequency distribution? Were the ToF values simply used onboard for removing agglomeration cases?

Section 2.5 Air flow Management:
- It looks strange that the air flow inside the instrument can be greater than the background air flow because the drag on the inside wall would reduce the air flow speed inside. Do you have some actual measurements showing this (see e.g., Appendix B of Fujiwara et al., 2016)?
- It is not clear why a double pendulum system would "inhibit the movement of UCASS". Is that due to a rather heavy radiosonde located below that acts as a drag? A double pendulum system might even give chaotic motions (depending on the mass of the second object and the air drag). Furthermore, in general, the payload may also move along a circle or ellipse in the horizontal plane.
- Related to the above two comments, in Section 1 or 2.1, an explanation is necessary why the instrument shape has been designed like this, i.e., not symmetric along the air flow; this would give additional complication here.
- In the end, all these factors go to the uncertainty of the measurements. The term "Management" in the section title may not be appropriate; more appropriate would be something like, Evaluation of the measurement uncertainty due to air flow uncertainty?

Section 3.1.1 Dropsonde system
- The date information for the sounding in Fig. 14 is necessary. (See also the major comment on this sounding.)
- What about the results from the other 5 soundings?

Section 3.1.2 Upsonde system
- Description on the PCASP onboard the research aircraft is necessary.

References:

Fujiwara, M., Sugidachi, T., Arai, T., Shimizu, K., Hayashi, M., Noma, Y., Kawagita, H., Sagara, K., Nakagawa, T., Okumura, S., Inai, Y., Shibata, T., Iwasaki, S., and Shimizu, A.: Development of a cloud particle sensor for radiosonde sounding, Atmos. Meas. Tech., 9, 5911-5931, https://doi.org/10.5194/amt-9-5911-2016, 2016.

Gao, R.S., Telg, H., McLaughlin, R.J., Ciciora, S.J., Watts, L.A., Richardson, M.S., Schwarz, J.P., Perring, A.E., Thornberry, T.D., Rollins, A.W., Markovic, M.Z., Bates, T.S., Johnson, J.E., and Fahey, D.W.: A light-weight, high-sensitivity particle spectrometer for PM2.5 aerosol measurements, Aerosol Science and Technology, 50, 88–99, https://doi.org/10.1080/02786826.2015.1131809, 2016.

Renard, J.-B., Dulac, F., Berthet, G., Lurton, T., Vignelles, D., Jégou, F., Tonnelier, T., Jeannot, M., Couté, B., Akiki, R., Verdier, N., Mallet, M., Gensdarmes, F., Charpentier, P., Mesmin, S., Duverger, V., Dupont, J.-C., Elias, T., Crenn, V., Sciare, J., Zieger, P., Salter, M., Roberts, T., Giacomoni, J., Gobbi, M., Hamonou, E., Olafsson, H., Dagsson-Waldhauserova, P., Camy-Peyret, C., Mazel, C., Décamps, T., Piringer, M., Surcin, J., and Daugeron, D.: LOAC: a small aerosol optical counter/sizer for ground-based and balloon measurements of the size distribution and nature of atmospheric particles – Part 1: Principle of measurements and instrument evaluation, Atmos. Meas. Tech., 9, 1721-1742, https://doi.org/10.5194/amt-9-1721-2016, 2016.

Masatomo Fujiwara

---

## Referee Comment (RC2) · Anonymous Referee #2 · 3 Apr 2019

Review of "The Universal Cloud and Aerosol Sounding System (UCASS): a low-cost miniature optical particle counter for use in dropsonde or balloon-borne sounding systems" by Smith et al.

This manuscript describes a low-cost miniature optical particle counter for relatively large particles. The design of the counter seems to be sound and unique. The manuscript is well written but there are a few issues that need to be addressed. I recommend significant revision before it can be accepted for this Journal. Detailed comments and suggestions are listed below.

[Figure]

Major comments:

1) Calibration process is incomplete. The described calibrations using particles of known diameters can only determine the sensitivities of the instrument, but not counting efficiencies (as a function of particle diameter). This is especially important for small particles (near the lower detection limit of 0.4 $\mu$m). Note that the inter-comparison results shown in Figs 16 and 17 are not sufficient to validate the UCASS. This is because the particle mass concentration or liquid water content is only sensitive to large particles. The authors need to demonstrate the counting efficiency by comparing the UCASS to a proven OPC or CN instrument.

2) It is a bit disturbing to see the wide (up to 1 order of magnitude) and inconsistent (between the low- and high-gain channels and between various sizes – note especially peaks of 0.753 and 3 $\mu$m PSLs and 11.58 $\mu$m soda lime) spreads of the instrument responses to PSL and other particles (Fig. 8). The authors seem to attribute it to calibration particles. But it is hard to believe the PSLs have such large spreads. If it is due to the real PSL spread, the authors ought to be able to reduce the spreads by using a DMA (at least for particles < 1 $\mu$m) and redo the calibration. If not due to the calibration material problem, then the authors need to provide an explanation.

3) If the large spreads shown in Fig. 8 are due to an instrument problem (such as the imperfection of sensing area definition/particle rejection as described in Fig. 4), then the size resolution of the instrument is not great. Detailed analysis is needed to show the true size resolution.

4) The description of the optical assembly is very difficult to understand. A better Fig. 2 should help.

Minor comments:

1) Fig. 1 is not well done. Appears to be hand drawn?

2) Fig. 16 needs to be improved.

3) It is hard to get a clear understanding of the electronics design. A circuit diagram should help (such as Fig 4. In Hill et al., J. of Atmos. And Ocean Tech., 2008).

4) A quick search for Alphasense mirror and First Sensor detector didn't yield any useful results. Please add web links or state that they special orders.

5) "f(x)" is not defined in Fig. 8.

---

## Referee Comment (RC3) · James Dorsey (Referee) · 12 Apr 2019

The manuscript describes a novel atmospheric particle spectrometer designed for use as part of a radiosonde / dropsonde system. The description and characterisation of the probe are comprehensive and require only minor improvements as described below. Field and laboratory based comparisons with more established particle measurement methods appear to show at least reasonable agreement, although the choice of particle mass for intercomparison appears at odds with the stated motivations for developing the system.

[Figure]

The manuscript is suitable for publication in AMT subject to some minor revisions. A large number of such changes are suggested, but these are all suggestions to improve an already good paper rather than requirements.

**Major comments**

Sections 3.1 and 3.2 show intercomparisons with a PCASP and a CDP respectively. The subject of this paper is an instrument which counts and sizes airborne particles, so it is confusing that the comparisons with other counting and sizing instruments is accomplished using data which has been converted to mass per unit volume. Especially given the motivations outlined in section 1 to improve understanding of aerosol radiative direct and indirect effects which are in large part controlled by particle number and surface area rather than mass. It can be difficult to compare three dimensional plots of size distribution as a function of time or height. However, it would be useful to see at least either a time series of particle number alongside the mass time series, or an averaged size distribution for UCASS and PCASP / UCASS and CDP in sections 3.1 and 3.2. This would improve confidence in the sample volume calculation outlined in section 2.5 as well as in the sizing accuracy of UCASS.

**Minor comments**

Section 2.1

Figure 2 uses a different but similar looking labeling system to figure 1. It might be worth numbering the parts in one of these diagrams, although this change is not essential. Figure 2 also appears to be less well drawn than the other figures in section 2.

It might be worth tidying it up.

Section 2.2

This section should possibly have some reference to dealing with coincidence errors or at least an estimate of the number concentration at which coincidence errors are likely to become significant.

Section 2.3

Line 3 on page 9 and subsequent parts of the paper contain references to 4095 bins of amplitude displacement. This is initially a bit confusing because in this context the output of a voltage converter as described is (very) often referred to as "Analogue to Digital Counts" or AD counts.

P9 L8 - It would be interesting to know why such a large range of particle time of flight is accepted.

P12 L10 Typo: none-turbulent should be non-turbulent. See also P14 L4 (none linearity), P19 L12 (none cloudy) and others.

Section 2.4.2

Presumably the sheath flow was added in order to accommodate the large volume of air flowing through the instrument. It would be useful to state this. It would also be interesting to know the length of the dryer column. A flow velocity of 5 m/s might not provide sufficient time to dry a flow containing PSLs using most conventional dryers.

On page 13 line 6 the authors discuss the use of PbP data to eliminate bin width related artefacts. They appear to be writing about exactly the same measures they

describe in section 2.3 (page 9 line 3), but using completely different terminology. This is confusing. PbP pulse height recording is a more widely understood terminology than that used in section 2.3 so it would be useful to standardise to this.

Figure 8 has f(x) as the Y axis label. This is normalised counts, but is not defined in the text or the figure legend.

Figure 9 on page 15 shows an additional step in the probe calibration relating scattering cross section to instrument response. It is more usual to see the calibration mode diameters plotted on top of the Mie curves as presented in figure 10. It would be useful to see the calibration added to figure 10 as well as (or even instead of) figure 9.

Section 2.5

More description about how the angles of oscillation were calculated would be interesting. Also, on line 2 of page 18 the authors give an airspeed of 5.4 +- 0.3 m/s. Reading the values for +-5 degrees from figure 11 seems to show a range of around 4.5 to 5.6 m/s. The authors should show how the former figure was arrived at.

Section 3

Figure 14 would be much easier to interpret if panels a - c used the same Y scale.

Section 3.1.2

The explanation of the differences between the UCASS and PCASP measurements sounds a little speculative. It raises a question about why these data are being used for an intercomparison if their imperfect colocation means they are not comparable. The agreement between the probes seems OK, so this could be left out (subject to the
major comment above being addressed).

P22 - Figure 16 appears to be at insufficiently high resolution or has been comressed using an excessively "lossy" method. Can this be re-plotted?

Section 3.2

Change "figure ??" to figure 17 on the first line of page 23.

The discussion on page 23 of the time of flight rejection causing under counting contains a mistake. The short time of flight of fast moving particles is rejected on the basis that it looks like short duration electronic noise, not on the basis that it looks like a large aggregate particle. At least accordin to the reasoning in section 2.3 (page 9).

Section 4

Line 8 of page 25 mentions the use of an 8+ point sizing calibration. Was this type of calibration applied to all probes contributing data in section 3, or was this done once as an instrument characterisation exercise?

---

## Author Comment (AC1) · 2 Jun 2019

Response to reviewer #1

responses are shown below reviewer comments

1. The instrument uses 658 nm laser light with an open-path configuration. Isn't there any influence of stray sunlight on the measurements? If not, is there any special measure for the daytime use?

[Figure]

response: The electronic circuit removes the background signal to account for stray light effects. In addition to this, the inside of the instrument is coated with an absorptive paint, preventing reflections down the inlet and therefore minimising the amount of stray light incident on the detector. In lab tests, these two methods have proved to eliminate counting/sizing errors due to stray light. This was not originally included in the manuscript for brevity, but will be amended in the revised version.

2. The authors wrote at page 9 that Time-of-Flight (ToF) data is used to "reject" signals that may come from "a large body or agglomeration of particles." Does this mean that such particle signals are not on the record? There is alternative approach in that such signals (including ToF data) are also recorded, processed, but may be removed in the data analysis phase. Is there any reason to reject those at the onboard circuit? I ask this question because this treatment may miss cloud signals if particles are greater than 40 $\mu$m. For example, in Figure 14, at 5 km, there is another $\sim$100 % relative humidity layer. It is not clear whether there was no cloud or there were clouds with particles much greater than 40 $\mu$m.

response: Due to limited bandwidth, we do not record particle-by-particle data, and only a subset of time-of-flight data is recorded for quality assurance. Therefore it is not possible to record all of this data and deal with it in the data analysis phase. The measurable size range is based solely on the pulse-height produced by the scattering particle, the pulse is digitised to a value between 1 and 4095 and so pulses too small or too high will not be recorded. Therefore, if the upper measurement limit of a unit is 40microns, then a particle larger than this will not be counted regardless of the time-of-flight. There is a relationship between the time-of-flight and particle size, which allows us to exclude erroneous data. i.e. if the upper measurable limit is 40microns, and given the ToF-size relationship, a particle with a pulse height (size) within the measurable range, but a time-of-flight much higher than the upper limit is likely to be some erroneous count. This could happen in the case of an agglomeration of particles. I will include some clarification in the revised manuscript

3. The description of the assembly, optical set-up, and sensing area is sometimes not easy to follow. I understand this because the instrument is three dimensional, but I have some ideas to improve this. I suggest the authors to (1) define the common x, y, and z axes for Figures 1 through 4, (2) show the axes explicitly in each of Figures 1 through 4, (3) avoid terminologies such as "width/depth", "left/right", "above/below" etc., but use the axis to specify the direction. Furthermore, please consider to use a common set of identifiers e.g., (a), (b), . . . in these four figures; for example, in Figure 2, (a)→(b1), (b)→(b2), (c)→(b3), (d)→(b4) (and keep Figure 1 as it is), and use e.g., (i), (ii), . . . instead of (a), (b), . . . in Figures 3 and 4. The mirror schematic in Figure 2 is very different from those in Figures 3 and 4, which made me confused at first. The laser light schematic in red in Figures 2, 3, and 4 may be improved by making them more consistent across these figures, and add an explanation in caption Figure 5 about how the "major axis" and "minor axis" correspond to the laser light schematic in Figures 2 through 4. Finally, please add the dimension information to Figures 1 through 3 as much as possible, so that it becomes easier to read the text by referring to these figures.

resonse: These are good suggestions and the diagrams will be amended accordingly. The mirror looks different in diagrams as sometimes a cross section is used and other times the entire unit is shown. I will stick with the cross section for all diagrams.

Section 1 Introduction: - I think MODIS and MISR are not lidars. –

response: I will amend this in the revised manuscript

There have been several particle instruments for balloon sounding. Some examples can be found in the Introduction of Fujiwara et al. (2016). Other instruments include LOAC (Renard et al., 2016) and POPS (Gao et al., 2016). - I think that an OPC for dropsonde may be new. I assume that the dimension, shape, and configuration of the UCASS was determined so that it can be mounted in the dropsonde launcher. If so, please explicitly write the conditions under which an instrument can be used as a

dropsonde. Also, the strength of the dropsonde over the balloon sonde system may be discussed (e.g., the former can be more easily targeted to a specific airmass including a specific cloud system). - Because the examples of aerosol particle measurements shown in this paper are actually only for Saharan mineral dust, the role of mineral dust particles on the climate may be explicitly discussed. - The mass of the instrument 280 g should be explicitly written here (near the term "lightweight"; and also in Section 2).

response: I will add in a discussion of balloon based instruments in the introduction. You are correct in that the shape of the UCASS was determined by the initial dropsonde design, in which the KITsonde had to fit entirely inside the UCASS, which in turn had to fit inside the drop tube. I will clarify this in the revised manuscript.

Section 2 Instrument Design, subsections 2.1 through 2.2: - See the major comments. - Why is the sensing area ("0.5 mm2") an area, not a volume? - It is helpful for the readers to summarize (e.g., to prepare a summary table for) the particle signal data (i.e., I1, I2, ToF, pulse height, etc.) and their relation with the criteria for data quality assurance (i.e., particle path inside/outside the sensing area, agglomeration of particles, etc.) Please also add the information how the numbers 0.4, 17, 1, and 40 $\mu$m for the particle size limits were actually determined. - Figures 3 and 4 indicate that the detector only collect light reflected on the mirror. There is no contribution from the directly scattered light (i.e., around 120 deg.)?

response: You are correct in that any particle passing within a particular volume will cause a pulse on the detector. However, we discuss a sensing area, as the airflow is perpendicular to the laser beam. Therefore, any particle passing through a particular AREA will travel through the depth of the beam. It is the area presented orthogonal to the airflow that dictates the sampling rate. The depth of the laser does not impact the sampling rate, it only affects the time-of-flight. I will clarify this in the revision, perhaps with a diagram.

Section 2.3 Electronics: Perhaps, the explanation of the "gain" i.e., "high gain version"

and "low gain version" is first described here at page 8. In the current manuscript, the gain is mentioned first in the Calibration section at page 13, and it is not very clear what the "gain" actually is.

response: We will add in some extra description where needed

Section 2.4.2 Calibration Measurements - Have you used all the calibration particles for each of the two different gain versions? In Figure 8, the set of particles shown is different between the two. For example, what is the response of the high-gain version to soda-lime 37.36 $\mu$m particles? - What were the ToF values for these experiments? How about the frequency distribution? Were the ToF values simply used onboard for removing agglomeration cases?

response: The high gain version has a smaller upper size limit. In each panel, the x-axis represents the full 4096 bins available. On the top panel, you can see that the measured size distribution for the 14.4micron calibration beads is cut off at the larger sizes, and the full distribution is not captured. Any pulses above 4095 cannot be counted. Therefore, the additional particles used for the low-gain calibration would not be seen at all by the high-gain version as they are off the scale. The ToF data during calibration is used in the same way that Tof data is used in experiments, where only overly low or high ToF values are rejected, corresponding to particle sizes below/above the measurable size range are rejected. I will clarify this in the revision

Section 2.5 Air flow Management: - It looks strange that the air flow inside the instrument can be greater than the background air flow because the drag on the inside wall would reduce the air flow speed inside. Do you have some actual measurements showing this (see e.g., Appendix B of Fujiwara et al., 2016)? - It is not clear why a double pendulum system would "inhibit the movement of UCASS". Is that due to a rather heavy radiosonde located below that acts as a drag? A double pendulum system might even give chaotic motions (depending on the mass of the second object and the air drag). Furthermore, in general, the payload may also move along a circle or ellipse in the

horizontal plane. - Related to the above two comments, in Section 1 or 2.1, an explanation is necessary why the instrument shape has been designed like this, i.e., not symmetric along the air flow; this would give additional complication here. - In the end, all these factors go to the uncertainty of the measurements. The term "Management" in the section title may not be appropriate; more appropriate would be something like, Evaluation of the measurement uncertainty due to air flow uncertainty?

response: You are correct in that the drag on the inside wall reduces the air speed in the boundary layer of the surface, however the sample area is outside this region and thus experiences a higher air flow. We can include measurements in the revision. Although a double pendulum is indeed chaotic, the maximum angle of tilt for the central element is very much limited in this configuration when compared to a single pendulum system. For the CFD modelling, we only showed the effects of tilt along one axis, because this is the only direction that has an impact on the airflow due to the location of the sample area close to one edge of the inlet. Along the other axis (I will define this in the revised manuscript using your suggestion of a universal coordinate system), the sample volume is sufficiently far away from the inlet edges, such that the tilt has no effect. I will try and explain this further in the revision, and will change the section title as suggested.

Section 3.1.1 Dropsonde system - The date information for the sounding in Fig. 14 is necessary. (See also the major comment on this sounding.) - What about the results from the other 5 soundings? I will add the date information.

response: Not all data is shown for brevity, the purpose of the paper is to give the technical information on the instrument. Some field data is included to show the various applications of the instrument (i.e. droplets/aerosol upsonde/dropsonde) but we don't think it is necessary to include all data from all campaigns. For the dropsonde tests, the other drops did not have comparative data available, or were tested in clear skies and so these data sets add little scientific value.

Section 3.1.2 Upsonde system - Description on the PCASP onboard the research aircraft is necessary.

response: I will add this.
* * *

---

## Author Comment (AC2) · 2 Jun 2019

responses are shown below reviwer comments

Major comments: 1) Calibration process is incomplete. The described calibrations using particles of known diameters can only determine the sensitivies of the instrument, but not counting efficiencies (as a function of particle diameter). This is especially important for small particles (near the lower detection limit of 0.4 $\mu$m). Note that the inter-comparison results shown in Figs 16 and 17 are not sufficient to validate the

UCASS. This is because the particle mass concentration or liquid water content is only sensitive to large particles. The authors need to demonstrate the counting efficiency by comparing the UCASS to a proven OPC or CN instrument.

response:The UCASS sensing area is defined optically, as described briefly in the paper. The hope with this is that the sensing area can be defined based on the fundamental geometry of the system, and therefore we do not have to rely on inter-comparisons with 'proven' instruments, which may have their own counting errors etc. The CDP was used for comparison because of the open geometry nature, meaning that it could be mounted alongside the UCASS and the results should be comparable. We have been unable to find an open path instrument to do the same comparisons for smaller sizes, but we do take your point and will try and take some comparative measurements in the lab. I will amend this in the revised manuscript.

2) It is a bit disturbing to see the wide (up to 1 order of magnitude) and inconsistent (between the low- and high-gain channels and between various sizes – note especially peaks of 0.753 and 3 $\mu$m PSLs and 11.58 $\mu$m soda lime) spreads of the instrument responses to PSL and other particles (Fig. 8). The authors seem to attribute it to calibration particles. But it is hard to believe the PSLs have such large spreads. If it is due to the real PSL spread, the authors ought to be able to reduce the spreads by using a DMA (at least for particles < 1 $\mu$m) and redo the calibration. If not due to the calibration material problem, then the authors need to provide an explanation.

response:The graph in question has the instrument response (amplitude displacement) on the x-axis. The instrument response does not correlate linearly with the particle size, so an order of magnitude in instrument response does not represent an order of magnitude in measured diameter. I will either alter this graph to show diameter rather than amplitude displacement, or add an additional graph to show this information. We had considered purchasing a DMA for experiments such as this, but they are only available for small size ranges (<5microns) and therefore cannot be used for a large number of our calibration particles, but we may look into this again. When taking size

calibrations, we check the size range against an APS and a TSI OPC to ensure we are producing particles in the correct size range. Perhaps in the absence of a DMA, we could show the measured size distributions by the UCASS alongside measurements from other instruments. Manufacturers also provide some statistical information for the calibration beads which can also be shown for comparison.

3) If the large spreads shown in Fig. 8 are due to an instrument problem (such as the imperfection of sensing area definition/particle rejection as described in Fig. 4), then the size resolution of the instrument is not great. Detailed analysis is needed to show the true size resolution.

response: As mentioned in the above comment, fig.8 does not show measured particle size, but instrument response. Therefore the spread in measured size is not so wide. The suggestions made in response to the above comment will be used to comment on size resolution.

4) The description of the optical assembly is very difficult to understand. A better Fig. 2 should help.

response: The optical assembly will be updated to follow a common coordinate system as suggested by reviewer #1, this should make the optical assembly easier to understand.

Minor comments:

1) Fig. 1 is not well done. Appears to be hand drawn?

response:Fig.1 is an automatically generated technical drawing from 3d modelling spftware, I will attempt to export it again to solve the issue with the edges.

2) Fig. 16 needs to be improved. C2

response: Fig 16 will be redone

3) It is hard to get a clear understanding of the electronics design. A circuit diagram

should help (such as Fig 4. In Hill et al., J. of Atmos. And Ocean Tech., 2008).

response: We may add in some more description or a simplified diagram to help with the explanation of the electronics system. However we will not include a full circuit diagram as this is commercially sensitive.

4) A quick search for Alphasense mirror and First Sensor detector didn't yield any useful results. Please add web links or state that they special orders.

response: These are special orders, I will clarify this in the manuscript.

5) "f(x)" is not defined in Fig. 8.

response: This is a probability density function, I will clarify this in the manuscript.

---

## Author Comment (AC3) · 2 Jun 2019

responses are shown under reviewer comments

Response to reviewer #3

Major comments Sections 3.1 and 3.2 show intercomparisons with a PCASP and a CDP respectively. The subject of this paper is an instrument which counts and sizes airborne particles, so it is confusing that the comparisons with other counting and sizing instruments is accomplished using data which has been converted to mass per unit

volume. Especially given the motivations outlined in section 1 to improve understanding of aerosol radiative direct and indirect effects which are in large part controlled by particle number and surface area rather than mass. It can be difficult to compare three dimensional plots of size distribution as a function of time or height. However, it would be useful to see at least either a time series of particle number alongside the mass time series, or an averaged size distribution for UCASS and PCASP / UCASS and CDP in sections 3.1 and 3.2. This would improve confidence in the sample volume calculation outlined in section 2.5 as well as in the sizing accuracy of UCASS.

response: Some of this comparative data is used simply because this was the data made available to us. But this is a good suggestion, I will have a look at the available data and try to implement this where possible.

Minor comments Section 2.1 Figure 2 uses a different but similar looking labeling system to figure 1. It might be worth numbering the parts in one of these diagrams, although this change is not essential. Figure 2 also appears to be less well drawn than the other figures in section 2. It might be worth tidying it up.

response: I will change the labelling system so that it is consistent for the whole section. I will also redo figure 2

Section 2.2 This section should possibly have some reference to dealing with coincidence errors or at least an estimate of the number concentration at which coincidence errors are likely to become significant.

response: I will add in an estimate for this

Section 2.3 Line 3 on page 9 and subsequent parts of the paper contain references to 4095 bins of amplitude displacement. This is initially a bit confusing because in this context the output of a voltage converter as described is (very) often referred to as "Analogue to Digital Counts" or AD counts. P9 L8 - It would be interesting to know why such a large range of particle time of fight is accepted. P12 L10 Typo: noneturbulent should be non-turbulent. See also P14 L4 (none linearity), P19 L12 (none cloudy) and others.

response: You may be correct about the AD meaning, I will check and correct it if necessary. Particle time of flight is related somewhat to the particle size, although there is realistically a large spread in ToF. The boundaries are set to remove unrealistic values of ToF based on empirical data. I may add more about this in the manuscript as reviewer #1 had some questions about ToF too. Typos will be corrected

Section 2.4.2 Presumably the sheath flow was added in order to accommodate the large volume of air flowing through the instrument. It would be useful to state this. It would also be interesting to know the length of the dryer column. A flow velocity of 5 m/s might not provide sufficient time to dry a flow containing PSLs using most conventional dryers. On page 13 line 6 the authors discuss the use of PbP data to eliminate bin width related artefacts. They appear to be writing about exactly the same measures they describe in section 2.3 (page 9 line 3), but using completely different terminology. This is confusing. PbP pulse height recording is a more widely understood terminology than that used in section 2.3 so it would be useful to standardise to this. Figure 8 has f(x) as the Y axis label. This is normalised counts, but is not defined in the text or the figure legend. Figure 9 on page 15 shows an additional step in the probe calibration relating scattering cross section to instrument response. It is more usual to see the calibration mode diameters plotted on top of the Mie curves as presented in figure 10. It would be useful to see the calibration added to figure 10 as well as (or even instead of) figure 9.

response: You are correct in that the sheath flow is used due to the large volume of air being used, it is essentially used to constrain particles withing the middle of the air flow allowing for fewer beads to be used. I will add in the length of the tube. The wet dispersed aerosol (PSL) are passed into the tube from another drying chamber before they enter the tube. I will clarify this in the revised manuscript. I will update the terminology as suggested to 'PbP pulse height recording' . f(x) is the probability density

function, I will define this in the text and caption. I think it is important to keep figure 9 to show the relationship with cross section, as this can be applied to any particle and is therefore useful in post calibrations. I will add the calibration particle information on figure 10 as suggested.

Section 2.5 More description about how the angles of oscillation were calculated would be interesting. Also, on line 2 of page 18 the authors give an airspeed of 5.4 +- 0.3 m/s. Reading the values for +-5 degrees from figure 11 seems to show a range of around 4.5 to 5.6 m/s. The authors should show how the former figure was arrived at.

response: The description of the pendulum system was not initially included for reasons of brevity, however this may be added as an appendix. The 'range' in airspeed is actually standard deviation, not a range, I will clarify this in the text (and also reconsider if this is the best way to describe the air flow).

Section 3 Figure 14 would be much easier to interpret if panels a - c used the same Y scale.

response: Good suggestion, I will correct this.

Section 3.1.2 The explanation of the differences between the UCASS and PCASP measurements sounds a little speculative. It raises a question about why these data are being used for an intercomparison if their imperfect colocation means they are not comparable. The agreement between the probes seems OK, so this could be left out major comment above being addressed). P22 - Figure 16 appears to be at insufiňĄciently high resolution or has been comressed using an excessively "lossy" method. Can this be re-plotted?

response: The use of field data is really intended to highlight the various uses of the UCASS and show (albeit speculatively) how it can be used to complement campaign datasets and potentially be used as an alternative where flights may not be possible. It is not intended as controlled proof of the instrument performance. Due to the additional

laboratory tests suggested by the other reviewers, I will try and make this clearer in the revised manuscript. i.e. laboratory data for intercomparison, field data for examples of use. Fig 16 will be redone.

Section 3.2 Change "fi̧gure ??" to fi̧gure 17 on the fi̧rst line of page 23. The discussion on page 23 of the time of flight rejection causing under counting contains a mistake. The short time of flight of fast moving particles is rejected on the basis that it looks like short duration electronic noise, not on the basis that it looks like a large aggregate particle. At least accordin to the reasoning in section 2.3 (page 9).

response: The 'figure ??' typo will be corrected. Also, the incorrect description of short ToF rejection will be rejected.

Section 4 Line 8 of page 25 mentions the use of an 8+ point sizing calibration. Was this type of calibration applied to all probes contributing data in section 3, or was this done once as an instrument characterisation exercise?

response: This was done as an instrument calibration exercise as more calibration standards have been made available since the first field tests were done. I will clarify this in the text and explain the calibration used for previous launches.

---

## Author Response (AR1)

Response to reviewer #1

1. The instrument uses 658 nm laser light with an open-path configuration. Isn't there any influence of stray sunlight on the measurements? If not, is there any special measure for the daytime use?

*I have added a section on stray light which describes a two fold approach to minimising and correcting for stray light*

2. The authors wrote at page 9 that Time-of-Flight (ToF) data is used to "reject" signals that may come from "a large body or agglomeration of particles." Does this mean that such particle signals are not on the record? There is alternative approach in that such signals (including ToF data) are also recorded, processed, but may be removed in the data analysis phase. Is there any reason to reject those at the onboard circuit? I ask this question because this treatment may miss cloud signals if particles are greater than 40 μm. For example, in Figure 14, at 5 km, there is another ~100 % relative humidity layer. It is not clear whether there was no cloud or there were clouds with particles much greater than 40 μm.

*In section ELECTRONICS, I have clarified how particles above/below the measurable size cannot be counted, and therefore how the choice of ToF rejection criteria is justified. i.e. if a 50um particle causes the detector to saturate, and wont be counted, then realistically, we don't need to extend the time-of-flight criteria to include very large particles as they cannot be counted anyway.*

3. The description of the assembly, optical set-up, and sensing area is sometimes not easy to follow. I understand this because the instrument is three dimensional, but I have some ideas to improve this. I suggest the authors to (1) define the common x, y, and z axes for Figures 1 through 4, (2) show the axes explicitly in each of Figures 1 through 4, (3) avoid terminologies such as "width/depth", "left/right", "above/below" etc., but use the axis to specify the direction. Furthermore, please consider to use a common set of identifiers e.g., (a), (b), … in these four figures; for example, in Figure 2, (a)⬜(b1), (b)⬜(b2), (c)⬜(b3), (d)⬜(b4) (and keep Figure 1 as it is), and use e.g., (i), (ii), … instead of (a), (b), … in Figures 3 and 4. The mirror schematic in Figure 2 is very different from those in Figures 3 and 4, which made me confused at first. The

laser light schematic in red in Figures 2, 3, and 4 may be improved by making them more consistent across these figures, and add an explanation in caption Figure 5 about how the "major axis" and "minor axis" correspond to the laser light schematic in Figures 2 through 4. Finally, please add the dimension information to Figures 1 through 3 as much as possible, so that it becomes easier to read the text by referring to these figures.

*Figures have been updated with your suggestions*

Section 1 Introduction:  - I think MODIS and MISR are not lidars.  –

*I will amend this in the revised manuscript*

*Changed to 'instruments' rather than lidar*

There have been several particle instruments for balloon sounding. Some examples can be found in the Introduction of Fujiwara et al. (2016). Other instruments include LOAC (Renard et al., 2016) and POPS (Gao et al., 2016). - I think that an OPC for dropsonde may be new. I assume that the dimension, shape, and configuration of the UCASS was determined so that it can be mounted in the dropsonde launcher. If so, please explicitly write the conditions under which an instrument can be used as a dropsonde. Also, the strength of the dropsonde over the balloon sonde system may be discussed (e.g., the former can be more easily targeted to a specific airmass including a specific cloud system). - Because the examples of aerosol particle measurements shown in this paper are actually only for Saharan mineral dust, the role of mineral dust particles on the climate may be explicitly discussed. - The mass of the instrument 280 g should be explicitly written here (near the term "lightweight"; and also in Section 2).

*I have added a brief description of other balloon based instruments at the end of the introduction section*

Section 2 Instrument Design, subsections 2.1 through 2.2: - See the major comments. - Why is the sensing area ("0.5 mm2") an area, not a volume? - It is helpful for the readers to summarize (e.g., to prepare a summary table for) the particle signal data (i.e., I1, I2, ToF, pulse height, etc.) and their relation with the criteria for data quality assurance (i.e., particle path inside/outside the sensing area, agglomeration of particles, etc.) Please also add the information how the numbers 0.4, 17, 1, and 40 µm for the particle size limits were actually determined. - Figures 3 and 4 indicate that the detector only collect light reflected on the mirror. There is no contribution from the directly scattered light (i.e., around 120 deg.)?

*You are correct in that any particle passing within a particular volume will cause a pulse on the detector. However, we discuss a sensing area, as the airflow is perpendicular to the laser beam. Therefore, any particle passing through a particular AREA will travel through the depth of the beam. It is the area presented orthogonal to the airflow that dictates the sampling rate. The depth of the laser does not impact the sampling rate, it only affects the time-of-flight. I have tried to clarify this further in the manuscript, perhaps the updated figures will help to illustrate it*

Section 2.3 Electronics: Perhaps, the explanation of the "gain" i.e., "high gain version" and "low gain version" is first described here at page 8. In the current manuscript, the gain is mentioned first in the

Calibration section at page 13, and it is not very clear what the "gain" actually is.

*A paragraph is added (2ⁿᵈ paragraph in electronics) to describe the gain and how.*

Section 2.4.2 Calibration Measurements - Have you used all the calibration particles for each of the two different gain versions? In Figure 8, the set of particles shown is different between the two. For example, what is the response of the high-gain version to soda-lime 37.36 µm particles? - What

were the ToF values for these experiments? How about the frequency distribution? Were the ToF values simply used onboard for removing agglomeration cases?

*The high gain version has a smaller upper size limit. In each panel, the x-axis represents the full 4096 bins available. On the top panel, you can see that the measured size distribution for the 14.4micron calibration beads is cut off at the larger sizes, and the full distribution is not captured. Any pulses above 4095 cannot be counted. Therefore, the additional particles used for the low-gain calibration would not be seen at all by the high-gain version as they are off the scale. The ToF data during calibration is used in the same way that Tof data is used in experiments, where only overly low or high ToF values are rejected, corresponding to particle sizes below/above the measurable size range are rejected. I will clarify this in the revision*

*I have added some explanation about the upper limits of the measurement, i.e. why the distributions are cut off at the top end because the particles are beyond the measurable size range. This should explain why the very large particles cannot be used to calibrate the high-gain version as they would not be measured at all.*

Section 2.5 Air flow Management: - It looks strange that the air flow inside the instrument can be greater than the background air flow because the drag on the inside wall would reduce the air flow speed inside. Do you have some actual measurements showing this (see e.g., Appendix B of Fujiwara et al., 2016)? - It is not clear why a double pendulum system would "inhibit the movement of UCASS". Is that due to a rather heavy radiosonde located below that acts as a drag? A double pendulum system might even give chaotic motions (depending on the mass of the second object and the air drag). Furthermore, in general, the payload may also move along a circle or ellipse in the horizontal plane. - Related to the above two comments, in Section 1 or 2.1, an explanation is necessary why the instrument shape has been designed like this, i.e., not symmetric along the air flow; this would give additional complication here. - In the end, all these factors go to the uncertainty of the measurements. The term "Management" in the section title may not be appropriate; more appropriate would be something like, Evaluation of the measurement uncertainty due to air flow uncertainty?

*I have added some additional description explaining why the air flow is higher, why the UCASS is asymmetric and also how the double pendulum, although chaotic, reduces the amplitude of oscillation.*

Section 3.1.1 Dropsonde system - The date information for the sounding in Fig. 14 is necessary. (See also the major comment on this sounding.) - What about the results from the other 5 soundings?

*Added the date. Not all data is shown for brevity, the selection of in-situ results are intended to illustrate potential uses for the UCASS and so instances are selected where comparative data exists. Not all dropsondes were launched during dust events, or at times with AERONET scans and therefore the data is of little interest to the reader. Wording has changed to focus on the 2 examples shown.*

Section 3.1.2 Upsonde system - Description on the PCASP onboard the research aircraft is necessary.

*Added in section 3.1.2*

*RESPONSE TO REVIEWER #2*

Major comments: 1) Calibration process is incomplete. The described calibrations using particles of known diameters can only determine the sensitivities of the instrument, but not counting efficiencies (as a function of particle diameter). This is especially important for small particles (near the lower detection limit of 0.4 µm). Note that the inter-comparison results shown in Figs 16 and 17 are not sufficient to validate the UCASS. This is because the particle mass concentration or liquid water content is only sensitive to large particles. The authors need to demonstrate the counting efficiency by comparing the UCASS to a proven OPC or CN instrument.

*A section 'counting comparisons' has been added, comparing UCASS concentrations to a TSI 3330*

2) It is a bit disturbing to see the wide (up to 1 order of magnitude) and inconsistent (between the low- and high-gain channels and between various sizes – note especially peaks of 0.753 and 3 µm PSLs and 11.58 µm soda lime) spreads of the instrument responses to PSL and other particles (Fig. 8). The authors seem to attribute it to calibration particles. But it is hard to believe the PSLs have such large spreads. If it is due to the real PSL spread, the authors ought to be able to reduce the spreads by using a DMA (at least for particles < 1 µm) and redo the calibration. If not due to the calibration material problem, then the authors need to provide an explanation.

*I have clarified in the manuscript that these graphs show instrument response and that this is not linearly proporational to size. A section has been added 'sizing comparisons' to better illustrate the sizing capabilities*

3) If the large spreads shown in Fig. 8 are due to an instrument problem (such as the imperfection of sensing area definition/particle rejection as described in Fig. 4), then the size resolution of the instrument is not great. Detailed analysis is needed to show the true size resolution.

*This has been addressed in the above comment*

4) The description of the optical assembly is very difficult to understand. A better Fig. 2 should help.

*Description and diagrams have been updated*

Minor comments:

1) Fig. 1 is not well done. Appears to be hand drawn?

*Figure has been redone*

2) Fig. 16 needs to be improved. C2

*Figure has been redone*

3) It is hard to get a clear understanding of the electronics design. A circuit diagram should help (such as Fig 4. In Hill et al., J. of Atmos. And Ocean Tech., 2008).

*A simplified box diagram has been added*

4) A quick search for Alphasense mirror and First Sensor detector didn't yield any useful results. Please add web links or state that they special orders.
   *Added 'available through special order'*

5) "f(x)" is not defined in Fig. 8.

*This is now defined*

RESPONSE TO REVIEWER #3

Major comments

Sections 3.1 and 3.2 show intercomparisons with a PCASP and a CDP respectively. The subject of this paper is an instrument which counts and sizes airborne particles, so it is confusing that the comparisons with other counting and sizing instruments is accomplished using data which has been converted to mass per unit volume. Especially given the motivations outlined in section 1 to improve understanding of aerosol radiative direct and indirect effects which are in large part controlled by particle number and surface area rather than mass. It can be difficult to compare three dimensional plots of size distribution as a function of time or height. However, it would be useful to see at least either a time series of particle number alongside the mass time series, or an averaged size distribution for UCASS and PCASP / UCASS and CDP in sections 3.1 and 3.2. This would improve confidence in the sample volume calculation outlined in section 2.5 as well as in the sizing accuracy of UCASS.

*More lab comparisons have been added to show the counting and sizing abilities of the UCASS, and therefore this section has been left as it is as it makes use of available data and illustrates the potential uses in the field.*

Minor comments

Section 2.1

Figure 2 uses a different but similar looking labeling system to figure 1. It might be worth numbering the parts in one of these diagrams, although this change is not essential. Figure 2 also appears to be less well drawn than the other figures in section 2. It might be worth tidying it up.

*Figures in this section have been redone to follow a universal coordinate system and set of identifiers*

Section 2.2

This section should possibly have some reference to dealing with coincidence errors or at least an estimate of the number concentration at which coincidence errors are likely to become significant.

*This is limited by the speed of the electronics, which become problematic before coincidence errors are expected, this is clarified in the 'electronics' section.*

Section 2.3

Line 3 on page 9 and subsequent parts of the paper contain references to 4095 bins of amplitude displacement. This is initially a bit confusing because in this context the output of a voltage converter as described is (very) often referred to as "Analogue to Digital Counts" or AD counts. P9 L8 - It would be interesting to know why such a large range of particle time of flight is accepted. P12 L10 Typo: none-turbulent should be non-turbulent. See also P14 L4 (none linearity), P19 L12 (none cloudy) and others.

*Nones changed to non, AD corrected*

Section 2.4.2

Presumably the sheath flow was added in order to accommodate the large volume of air flowing through the instrument. It would be useful to state this. It would also be interesting to know the length of the dryer column. A flow velocity of 5 m/s might not provide sufficient time to dry a flow containing PSLs using most conventional dryers. On page 13 line 6 the authors discuss the use of PbP data to eliminate bin width related artefacts. They appear to be writing about exactly the same measures they describe in section 2.3 (page 9 line 3), but using completely different terminology. This is confusing. PbP pulse height recording is a more widely understood terminology than that used in section 2.3 so it would be useful to standardise to this. Figure 8 has f(x) as the Y axis label. This is normalised counts, but is not defined in the text or the figure legend. Figure 9 on page 15 shows an additional step in the probe calibration relating scattering cross section to instrument response. It is more usual to see the calibration mode diameters plotted on top of the Mie curves as presented in figure 10. It would be useful to see the calibration added to figure 10 as well as (or even instead of) figure 9.

*I have stated the reason for the sheath flow*

*Wet samples are passed through a drying chamber first, have changed this in the text to state that this drying column AIDS the drying, therefore suitable for dry dispersion, or almost dry wet samples. Section 2.3 changed to used PbP terminology. F(x) defined in caption. Data points added to calibration curves as suggested*

Section 2.5

More description about how the angles of oscillation were calculated would be interesting. Also, on line 2 of page 18 the authors give an airspeed of 5.4 +- 0.3 m/s. Reading the values for +-5 degrees from figure 11 seems to show a range of around 4.5 to 5.6 m/s. The authors should show how the former figure was arrived at.

*Changed this to mean and SD.*

Section 3.1.2

The explanation of the differences between the UCASS and PCASP measurements sounds a little speculative. It raises a question about why these data are being used for an intercomparison if their imperfect colocation means they are not comparable. The agreement between the probes seems OK, so this could be left out major comment above being addressed). P22 - Figure 16 appears to be at insufficiently high resolution or has been comressed using an excessively "lossy" method. Can this be re-plotted?

*The use of field data is really intended to highlight the various uses of the UCASS and show (albeit speculatively) how it can be used to complement campaign datasets and potentially be used as an alternative where flights may not be possible.  It is not intended as controlled proof of the instrument performance.  Due to the additional laboratory tests included in this version of the manuscript, this section has been left as it is*

Section 3.2

Change "figure ??" to figure 17 on the first line of page 23. The discussion on page 23 of the time of flight rejection causing under counting contains a mistake. The short time of flight of fast moving particles is rejected on the basis that it looks like short duration electronic noise, not on the basis that it looks like a large aggregate particle. At least accordin to the reasoning in section 2.3 (page 9).

*Typo corrected*

*ToF corrected*

Section 4

Line 8 of page 25 mentions the use of an 8+ point sizing calibration. Was this type of calibration applied to all probes contributing data in section 3, or was this done once as an instrument characterisation exercise?

*I have added that legacy measurements were done using a rougher calibration prior to the availability of borosilicate standards.*